



# Twelve years profile soil moisture and temperature measurements in Twente, the Netherlands

Rogier van der Velde, Harm-Jan F. Benninga, Bas Retsios

Department of Water Resources, Faculty of ITC, University of Twente, Enschede, 7500 AE, The Netherlands

*Correspondence to*: Rogier van der Velde (r.vandervelde@utwente.nl)

**Abstract.**

Spread across the Twente and neighbouring regions in the east of the Netherlands, a network of 20 profile soil moisture and temperature (5 cm, 10 cm, 20 cm, 40 cm and 80 cm) monitoring stations has been operational since 2009. In addition, field campaigns have been conducted covering the growing seasons of 2009, 2015, 2016 and 2017 during which soil sampling rings

and handheld probes were used to measure the soil moisture content of in total 28 fields near 12 different monitoring stations. In this paper, we describe the design of the monitoring network and the field campaigns, adopted instrumentation, experimental setup, field sampling strategies, and the development of sensor calibration functions. The maintenance and quality control procedures, known issues specific to the Twente network, and time series of profile soil moisture and groundwater level for three stations are discussed. Further, an overview is provided of open third-party datasets (i.e. land cover/use, soil information,

elevation, groundwater and meteorological observations) that can support the use and analysis of the Twente soil moisture and temperature datasets beyond the scope of this contribution. The data discussed are publicly available at https://doi.org/10.17026/dans-xr2-m6d8 (Van der Velde and Benninga, 2022) under the Creative Commons, CC BY 4.0 license.

## 1 Introduction

In virtually every hydrology textbook (e.g. Maidment 1993, Dingman 1993, Brutsaert 2005) one can read that water in the unsaturated soil, hereafter soil moisture, is needed for plants to grow, for groundwater to recharge, and for determining whether rain infiltrates or runs off laterally and contributes to the production of streamflow. Moreover, the conversion of water into vapour via evapotranspiration at the expense of solar radiation links soil moisture to the atmosphere impacting weather and climate (Seneviratne et al. 2010). Since its foundation in 1992, the Global Climate Observing System (GCOS) acknowledges

the crucial role soil moisture plays in the Earth's climate system, supports the development of long term global monitoring programmes (GCOS, 2004) and has recognized soil moisture as an essential climate variable (GCOS 2010). Considerable developments have taken place in global soil moisture monitoring with the launch of dedicated microwave satellites, e.g. Soil Moisture and Ocean Salinity (SMOS, Mecklenburg et al. 2016), Soil Moisture Active Passive (SMAP, Entekhabi et al. 2010) and long-term satellite based data products have become available (Gruber et al. 2019). In addition, an International Soil





Moisture Network (ISMN) has been established that hosts in situ soil moisture measurements from across the globe (Dorigo et al. 2011, 2021).

The number of in situ soil moisture monitoring programmes, dating back to the 1930s, has been small and many relied on gravimetric soil sampling (Robock et al. 2000). Gravimetrically determined soil moisture measurements are, however, labor intensive and have become unfeasible for long-term monitoring as the cost of labor increased. Therefore, indirect estimation

of the soil water content has been widely investigated (e.g. Vereecken et al. 2008), of which the devices that measure the soil's relative electric permittivity have become the commonly used instruments to base regional scale soil moisture monitoring networks on (e.g. Martinez-Fernandez and Cebalos 2005, Calvet et al. 2007, Su et al. 2011, Bircher et al. 2012, Smith et al. 2012, Benninga et al. 2018, Bogena et al. 2018, Caldwell et al. 2019, Tetlock et al. 2019). Despite that technological advances facilitated a substantial increase in the in-situ soil moisture monitoring infrastructure, in situ monitoring networks providing

long-term and consistent soil moisture data records are still very scarce across the globe (GCOS, 2016).

In this manuscript, we report on in situ profile soil moisture and soil temperature datasets collected by a regional scale monitoring network composed of 20 measurement locations that has been operational in the Twente region situated in the east of the Netherlands. Development of the Twente network began in the fall of 2008 and was completed by the summer of 2009, and has been operational ever since. Dente et al. (2011) described the early development and the first scientific use of the data

was the validation of SMOS soil moisture products (Dente et al. 2012). Other studies performed with the datasets have focused on field scale soil moisture retrieval (Van der Velde et al. 2015, Benninga et al. 2020), upscaling of point measurements to coarse satellite footprints (Van der Velde et al. 2021), agricultural and hydrological applications (Carranza et al. 2018, 2019, Pezij et al. 2019, Buitink et al. 2020) and the Twente network has been used as one of the core international validation sites for the SMAP surface soil moisture products (Colliander et al. 2017, Chan et al. 2018 Chaubell et al. 2020).

Over the years the design of the monitoring network has been impacted by gradual changes, which have not been reported in a consistent manner before. The manuscript provides an overview of the network developments, and describes the availability and processing of the collected datasets, including the calibration, maintenance, and quality control, procedures. In addition to the measurements collected in an automated and continuous manner, the dataset discloses the soil moisture records obtained with handheld probes and through gravimetric soil sampling during field campaigns conducted in 2009, 2015, 2016 and 2017.

Further, we describe open third-party datasets that can support the use of the soil moisture data, which include maps of land cover and use, maps of soil type, texture and physical properties, high resolution (0.5 m and 5.0 m) digital terrain models (DTM), groundwater level measurements, and meteorological observations

## 2 Study area and open datasets

### 2.1. Regional characteristics and water governance

Twente, about 1500 km$^2$, is a region in the Netherlands directly bordering Germany towards the east and bound in the west by a glacial ridge known as the Sallandse Heuvelrug. The majority of the network is situated in Twente, other parts are located in



the neighboring regions Salland and Achterhoek with similar characteristics. Glacial ridges formed in the second last glaciation period (Saalien) define the landscape. They have maximum elevations of around 80 m above mean sea level (a.m.s.l.) and consist mostly of fluvial sand deposits with glacial boulder clay sheets. This geomorphological feature in combination with a

temperate oceanic climate (*Cfb* Köppen-Geiger climate classification; Beck et al. 2018) facilitated the development of a drainage system composed of brooks and small unnavigable rivers flowing via larger rivers into the IJssel lake. Although deeper groundwater levels of 6 m up to 10 m below the surface can be found on the glacial ridges, they are generally shallow and fluctuate from within the top 1 m of soil layer during winters up to maximum depths of 2 m – 3 m in summers.

In the Netherlands, Twente and surroundings are considered rural areas with a few mid-sized and small cities, and a number

of villages, and are known for their characteristic bocage landscape with small agricultural fields (1.63 hectares on average) separated by tree lines and bushes amidst gently rolling topography. The majority of the agriculture has a focus on animal husbandry, whereby the available land is used to produce food for livestock via grass meadows and the growing of maize.

Three types of public institutions are mandated with the regional governance of water in the study area as described in the Water Act 2009 and the Regional Water Authorities Act 1991. They are provinces, regional water authorities (RWAs) and

municipalities. The provinces hold responsibility over the regional groundwater systems. The RWAs are accountable for the water quality and quantity in regional surface water systems, and the municipalities are charged with the urban rainwater collection and groundwater. Readers are referred to Havekes et al. (2017) for more details. Twente is part of the province Overijssel and falls under the RWA Vechtstromen. Figure 1 shows, however, that the network also covers the northern part of the province Gelderland, and extends towards the west to RWA Drents Overijsselse Delta and towards the south to RWA Rijn

and IJssel.

A large number of basic datasets are freely available for the Netherlands through various initiatives. In the following sections, we describe datasets on topography, soil, groundwater, land cover and weather that can support the use of the Twente soil moisture and temperature dataset. Section 6 describes how these datasets can be accessed.

## 2.2 Topography, soils and groundwater

Detailed spatial elevation data is available from the AHN ('Actueel Hoogtebestand Nederland' in Dutch). AHN (2019) supplies 0.05 m accurate and high resolution DTMs obtained via airborne laser altimetry. In 2019, the third version (AHN3) has been completed and made available with spatial resolutions of 0.5 m and 5.0 m. The DTM for the area covered by the monitoring stations is shown in Fig. 2 along with the various locations of the monitoring stations throughout the observation period.

Soil information up to a depth of 1.2 m can be obtained from the soil physical units map of the Netherlands named BOFEK

('BOdemfysische Eenheden Kaart' in Dutch). BOFEK combines the soil map of the Netherlands with the Dutch class pedotransfer function (Heinen et al. 2021). A subset of the soil type map for the study area is shown in Fig. 3.

The Netherlands have a comprehensive network of groundwater monitoring wells supported by various public organisations all contributing to a central database that is disseminated via DINOLoket ('Data en Informatie van de Nederlandse Ondergrond' in Dutch) managed by the Geological Survey of the Netherlands (2021). Not all monitoring wells in the database have records



that cover the observation period of the Twente network. We have, therefore, selected wells nearest to our monitoring stations with matching data coverage and shown them as points in Fig. 3. The well ID, coordinates, and distance to the associated soil moisture stations are listed in supplement Table S1.

The DTM of Fig. 2 shows that the study area has little relief sloping gently from about 5 m a.m.s.l. in the west to 30 m a.m.s.l. in the east, with some glacial ridges up to 80 m a.m.s.l.. The soil map in Fig. 3 shows that sand is the major soil type. On the

eastern glacial ridge also wind-blown loamy deposits can be found near the surface. Organic and peaty soils are present in the parts where water naturally stagnates, which has been the case for a major area in the north and along streams where also clayey soils exist.

## 2.3 Land cover

Land use information is publicly available from Statistics Netherlands and from the Ministry of Economic Affairs and Climate

Policy. Statistics Netherlands (2015) provides the main land use classes based on an interpretation of a 1:10.000 topographic map of the Netherlands and is published every two to four years since 1989. The Ministry of Economic Affairs and Climate Policy (2021) is responsible for the crop parcel registry. Since 2009, every land owner in the Netherlands has to report each year the crop on each parcel in their possession.

Figure 4 shows the 2015 land use map from Statistics Netherlands for the study area, from which can be deduced that 70.2 %

of the land is used for agricultural activities, 13 % is woodland, 11.3 % is built-up and the remaining 5.5 % is classified as water, recreational, dry and wet nature. The map illustrates that the larger forested areas are mainly found on the elevated glacial ridges and the agricultural activities take mostly place on the post-glacial soils covering the glacial pelvis. From the crop parcel registry in 2015, we find that the agricultural land is covered for 70.8% by grass meadows, 22.4 % by maize and the remaining 6.8% is used for potato, cereals, forest, heath, and other crops. The grass growing season is generally from

March till November during which the meadows are either being grazed by cattle or cut four to six times per year (Benninga et al., in preparation, 2021). Maize is planted in the months April/May and harvested in October/November depending on the vehicle bearing capacity of the land and growing conditions.

## 2.4. Climate and weather

In Fig. 4 the locations of the 3 automated weather and 29 precipitation stations operated by the Royal Netherlands

Meteorological Institute ('Koninklijk Nederlands Meteorologisch Instituut' in Dutch; KNMI 2021) in the study area are shown. The precipitation stations are part of a network of more than 300 voluntary observers in the Netherlands. The observers record manually with a 0.1 mm resolution the rainfall collected with a World Meteorological Organization (WMO) standard gauge around 9:00 CET in the morning and measure the snow depth with ruler when applicable. The data sent to the KNMI are validated in 10 day blocks and made available as daily values. The three automated weather stations are situated near the

villages Heino and Hupsel, and at Twenthe airport nearby Enschede. They measure wind speed and direction, air temperature at 1.5 m and 0.1 m above the surface, sunshine duration, shortwave incoming radiation, precipitation, air pressure, humidity,



and cloud cover. The adopted instrumentation and measurement protocols are according to international standards, and the quality controlled data are available as hourly and daily values. The daily set also holds the reference crop evapotranspiration ($E_{ref}$) calculated through application of the modified Makkink method described in De Bruin (1987).

Figure 5 shows for the period 2008 – 2020 the monthly average of daily mean 1.5 m air temperature as well as monthly precipitation and $E_{ref}$ sums, derived as mean values for the three automated weather stations. The data in this figure confirms that the soil moisture monitoring network is located in a temperate oceanic climate zone ($Cfb$). The coldest and warmest months have been January and July with mean monthly temperatures of 2.9 °C and 18.3 °C, respectively. Precipitation has been evenly distributed throughout the year according to the Köppen-Geiger classification, even though a difference of 53.3 mm exists in

sums between the driest (April, 33.5 mm) and wettest (August, 86.8 mm) month.

In the past fourteen years, the annual precipitation and $E_{ref}$ sums available for the three weather stations have been on average 757.1 mm and 611.3 mm, respectively, resulting in an annual surplus of 145.8 mm. In the years 2018, 2019 and 2020 north-western Europe has been struck by droughts (e.g. Buitink et al., 2020; Bakke et al., 2020; Buras et al., 2020) with less than normal precipitation volumes and higher evaporative demands. The most extreme rain day occurred on 26 August 2010, with

49.6 mm, 142.3 mm and 106.4 mm collected at KNMI stations Heino, Hupsel and Twenthe. The second and third heaviest rain days took place on 10 September 2013 with 22.3 mm, 74.5 mm and 57.8 mm and 3 August 2011 with 55.6 mm, 24.9 mm and 20.6 mm at Heino, Hupsel and Twenthe stations, respectively, while on all the other days less than 50 mm was recorded.

### 3. Monitoring network

#### 3.1 Sites

The development of the soil moisture and temperature monitoring network started in November 2008 and was completed in November 2009, but 19 out of the 20 stations were installed already before July 2009. The prime objective for the development of the measurement infrastructure was to serve as reference for the validation and calibration of coarse resolution soil moisture products derived from active and passive microwave satellite observations (Dente et al. 2011). The measurement sites are spread over a roughly 45 km x 40 km area and the individual stations are typically 5 km to 13 km apart, see also Fig. 2.

In the site selection care was taken to evenly distribute across the land covers and soil types. The majority of stations are found on sandy soils, two stations have been installed in sandy soils with a higher organic matter content, one in a loamy soil and one in a clayey soil according to the BOFEK soil map. The land on which the monitoring took place is in all cases privately owned and actively used for farming. The instrumentation is, therefore, typically placed at the border of fields and preferably several tens of metres away from disturbing features (i.e. trees, roads or watercourses), as shown in Fig. 6, to minimize

disturbance from recurring farming practices and optimize its representativeness for the adjacent fields.

Since the completion of its development, the monitoring network has been constantly subject to modifications, such as land cover changes as a result of crop rotation, and re-installations due to changes in land ownership or equipment failures. Table 1 lists for each station the main soil type as indicated in the soil map, the land cover per year of the adjacent fields and the





maintenance operations carried out. The location of the stations and their installation date are available as a list of geographic
(datum: WGS84) and map projected (Amersfoort/RD New, EPSG: 28892) coordinates.

## 3.2 Instrumentation and measurement setup

The Twente soil moisture and temperature monitoring network is built with instrumentation manufactured by METER Group
(formerly: Decagon Devices). The standard and remote versions of EM50 data logger series have been deployed to perform
measurements every minute with ECH$_2$O EC-TM and 5TM (firmware versions 2013 and 4.0) probes, and were set to record
readings at 15 minute intervals. Equipment of METER Group devices has previously been used for the development of many
monitoring networks, such as HOBE in Denmark (Bircher et al. 2012), TERENO in Germany (Bogena et al. 2018) and the
Raam in the Netherlands (Benninga et al. 2018), and been evaluated in several intercomparison studies (e.g. Jackisch et al.
2020, Vaz et al. 2013, Robinson et al. 2008).

The ECH$_2$O TM probes have a total length of 10.9 cm, a width of 3.4 cm and consist of a coated circuit board with an oscillator
that applies a 70 MHz electromagnetic wave to three 5 cm long fiberglass enclosed prongs. The prongs are 5 mm wide and 1
mm thick and are placed 5 mm apart. A thermistor near one of the prongs measures the temperature with an expected +/- 1ºC
accuracy and 0.1 ºC resolution. Probes estimate the volumetric soil moisture (VSM) by characterizing the apparent relative
electric permittivity via measurements of the capacitance, quantified as the charge needed to polarise the dielectric (soil)
surrounding the prongs (Decagon Devices 2008 and 2017). In Benninga et al. (2018), we have shown under laboratory
circumstances that the influence zone of 5TM probe in a sandy soil is around 3 cm to 4 cm.

Figure 6 illustrates typical measurement setups of the Twente network with probes installed at nominal depths of 5 cm, 10 cm,
20 cm, 40 cm and 80 cm. However, due to budget constraints several stations are limited to the upper two, three or four
measurement depths. At sites with a permanent grass cover, excavation of the installation pit started with cutting the grass sod
of an area of approximately 40 cm by 40 cm after which the top 10 cm to 15 cm (soil layer including grass) was carefully
removed and the pit was dug further until the required depth. The probes were installed in a lateral direction with the small
sides of the prongs pointing upward to avoid water ponding on the prongs, and with the printed text on the prongs in the upright
direction to ensure consistency in the depth of the thermistor. After installation the pit was back filled while compacting the
soil several times during the filling process, the grass sod was placed back and a trench was dug to guide the cables to a pole
on which the EM50 logger was mounted. The excess cables were buried near the pole. Typically a few months after installation
the plot would have returned to its original land cover. A similar installation procedure was adopted for cultivated land.

## 3.3 Capacitance probe calibration

Estimation of the VSM using the capacitance technique relies on the contrast between the relative electric permittivities ($\varepsilon_r$) of
air (1), soil (2-7) and water (80). Soil-specific calibrations are needed for two main reasons: i) to account for losses (imaginary
component of $\varepsilon_r$) due to the molecular relaxation and electric conductivity that alter the $\varepsilon_r$ as it appears to a capacitance sensor
(Robinson et al. 2008) and ii) the soil dependent dielectric response to VSM. Hence, soil-specific calibration functions have



been developed for both EC-TM and 5TM probes in the laboratory following the guidelines recommended by the manufacturer (Cobos and Chambers 2010). With this approach we assume that the sensor-to-sensor variability is accounted for by the in house calibration performed at the manufacturer against known standards. This can be justified based on the small variability (0.01 m³ m⁻³) among sensors evaluated by Kizito et al. (2008) and Rosenbaum et al. (2010).

In Dente et al. (2011) the development of the calibration function for the EC-TM probe is described. They performed the calibration on soil collected from 10 sites and could identify three relationships, but at the same time could not attribute this to a specific soil feature. Therefore, the recommendation was to use a generalized calibration function, expressed by

$$\theta_{cp} = a + b\theta_p, \tag{1}$$

where $\theta$ stands for the VSM (m³ m⁻³), $a$ and $b$ are the intercept (m³ m⁻³) and slope (-) of the linear regression function, and subscripts p and cp indicate the native probe reading and calibrated probe value. The native probe reading is a direct sensor

output obtained by applying the mineral soil calibration to the raw signal (Decagon Devices, 2008). Dente et al. (2011) report an $a$ of 0.0706 m³ m⁻³ and $b$ of 0.7751 yielding a root mean square error (RMSE) of 0.023 m³ m⁻³.

The calibration for the 5TM probe was performed in 2015 for soil taken from three sites each belonging to one of three groups earlier identified in Dente et al. (2011). The selected sites were ITC_SM03, ITC_SM07 and ITC_SM08. Similar as for the calibration of the EC-TM probe, soil was taken from the field in an ordinary 12 L bucket and was air dried. The air dried soil

was gradually wetted by adding 50 – 75 ml water at a time and after careful mixing a 5TM reading and 100 cm³ soil sample was taken. The soil sample was used to determine the VSM from the difference in wet and dry weight of the sample after 24 hours in the oven at 105 °C, which is referred to as gravimetrically determined volumetric soil moisture (GVSM). The entire process was done twice and resulted in 38 matchups for ITC_SM03, 32 for ITC_SM07 and 29 for ITC_SM08.

Figure 7a shows the GVSM against the 5TM VSM. Linear equations of the same type as Eq. (1) were fitted through the

matchups for each soil individually and all together. Because of the small sample size, the linear fits have been carried out for each combination of entire collections minus one (n-1). The matchup left out of the regression is then used for validation and the calculation of the performance metrics.

Table 2 lists the linear regression coefficients ($a$ and $b$) obtained for the four sets of matchups along with the standard deviation ($\sigma$) computed from the collection of regression coefficients for each individual set. The RMSE and mean error (ME) calculated

from the matchups left for validation and the coefficient of determination ($R^2$) obtained with the mean regression coefficients are provided as well. The listed metrics demonstrate that the performance of the 5TM sensor is in line with that of the EC-TM given the negligible MEs, RMSEs varying from 0.024 m³ m⁻³ to 0.031 m³ m⁻³ and $R^2$s in the 0.79-0.93 range. Even though the regression coefficients differ among the analysed soils, their point clouds in Fig. 7a have quite some overlap, which does not justify using different calibration functions. This is further supported by the fact that the $\sigma$ is only a fraction of the magnitude

of the regression coefficients when including all matchups. Notably, the obtained $\sigma$s are 4.8 % of the intercept and less than 0.5 % of slope relative to the magnitude, while it goes up to a respective 44 % and 2.4 % when using data from a single site. This suggests that the reliability of the function fitted through all matchups is higher. Therefore, we have chosen to apply the 'all soils' calibration function to every site of the Twente network, which is expected to provide an accuracy (RMSE) of 0.028



$m^3$ $m^{-3}$. Figure 7b presents the validation with the GVSM plotted against the 5TM VSM using the 'all soils' mean regression
coefficients.

## 4 Field campaigns

Field campaigns were conducted in 2009, 2015, 2016 and 2017, during which soil moisture was measured in fields with
handheld impedance probes and via soil samples taken for GVSM determination. The sampling took place at a maximum of
three fields owned by the same farmer adjacent to or near the monitoring station. This resulted in a total of 28 sampled fields
near 12 monitoring stations.

The general concept of each field campaign was similar, yet the execution differed every year. For instance, sampling days in
2009 and 2015 took place weekly from the end of summer in September until the beginning of November. In 2016 and 2017,
the sampling days were held weekly or biweekly depending on weather and staff availability, and covered the entire growing
season from April/May till the end of fall in November. An overview of the field campaigns is provided in Table 3, which
includes the time period, the number of sampling days and the sampled stations. The following sections describe the sampling
strategy, the instrumentation and the calibration of the probe readings.

### 4.1 Sampling strategy

The sampling strategy during campaigns aimed at characterizing the top 5 cm soil moisture content of fields. Three to six
locations, about 50 m to 100 m apart, were selected within the field to perform the measurements, depending on the parcel
size. As an example, Fig. 8 shows the scheme applied around ITCSM_02 for fieldwork conducted in 2016 and 2017.

Figure 9 illustrates the sampling strategy at sampling points. The number of handheld impedance probe readings per sampling
point varied from nine in the 2009 field campaign to five readings in 2015 and four in 2016-2017. At grass-covered fields, soil
moisture was measured with the impedance probe at four to nine points within a 1 $m^2$ plot and next to one of the probe readings
a soil sample was taken for GVSM determination. In maize fields, probe readings were taken along the transect perpendicular
to the crop rows, approximately 0.75 m apart, with the soil sample taken in between two rows. The collection of soil samples
for GVSM determination was done to calibrate the probe readings and stopped when the covered dynamic range and number
of matchups were suitable to establish a calibration function. We have noted in the provided data sheet which probe reading
corresponds to the GVSM.

### 4.2 ThetaProbe and HydraProbe

The Delta-T ThetaProbe (Type ML2; Delta-T Devices, 1998) and Stevens HydraProbe (analog version; Stevens Water
Monitoring Systems, 2020) are the two handheld probes that were used for rapid soil moisture data collection during the field
campaigns. Both instruments exploit the impedance mismatch between a coaxial transmission and a stainless steel pin inserted
in the soil that acts as a waveguide and is electrically shielded by three other similar pins (60 mm and 57 mm in length,





respectively) placed in an equilateral triangle with sides of 26.5 mm and 22 mm, respectively (Seyfried and Murdock 2004).

The ThetaProbe measures the amplitude difference of a standing sinusoidal wave between the start of a transmission line and the junction where the pins enter the soil as a result of the applied 100 MHz signal. The amplitude difference is used to determine the impedance from which the apparent relative electric permittivity is derived (Gaskin and Miller, 1996). The HydraProbe measures the complex ratio of the reflected and incident voltage of an applied 50 MHz signal to characterize the impedance of the soil to determine the complex relative electric permittivity (Campbell 1990, Kraft 1987). Both the ThetaProbe and HydraProbe data loggers have built-in software to convert the voltage output to a soil moisture content. In addition to soil moisture, the HydraProbe also provides bulk electric conductivity and temperature. Because the relationship between $\varepsilon_r$ and VSM is affected by the soil type, calibration of impedance probe measurements is generally needed. In case of the ThetaProbe, the calibration accounts also for conductive and molecular losses, which is less of an issue with the HydraProbe as it measures independently the real and imaginary components of the relative electric permittivity.

### 4.3 Impedance probe calibration

The measurements of the 2009 and 2015 field campaigns were collected with the ThetaProbe, during which a total of 93 and 166 matchups with GVSM were collected at fields near eight and six different stations, respectively. Figure 10 presents plots of GVSM against the ThetaProbe VSM with in the upper panels (Figs. 10a and 10b) the 2009 data and in the lower panels (Figs. 10c and 10d) the 2015 data. The GVSM against the matching ThetaProbe readings is shown in Figs 10a and 10c, and the GVSM against the mean of the readings at a sampling point is shown in Figs. 10b and 10d.

In general, it can be noted that all plots show positive relationships and that the scatter among the data points is clearly less in 2015 in comparison to 2009. This is particularly the case for the matching ThetaProbe readings. The explanation for this difference in performance between the years is a combination of the larger number of stations sampled in 2009, the lower number of matchups available for 2009, and also the operator's skills could have played a role. Regardless of the scatter noted in the data points of 2009, it is difficult to identify distinct relationships for individual stations. Among the 2015 data points clusters belonging to a single station are observed, but this is primarily due to the persistent soil moisture levels at specific stations. The attribution of a GVSM – ThetaProbe relationship to a specific soil type or station remains unclear. Therefore, we have chosen to develop the calibration functions for the ThetaProbe on a field campaign basis and not to make a distinction between individual stations. This also ensures a sufficient number of matchups and a larger soil moisture range.

The data collection of the 2016 and 2017 field campaigns was performed with the HydraProbe and took place near three stations (ITC_SM02, ITC_SM07, and ITC_SM10) in 2016, to which ITC_SM03 was added in 2017. A total of 285 pairs of GVSM and HydraProbe readings were acquired, with > 86 matchups for each station at which the measurements started in 2016 and 12 matchups for ITC_SM03. Figure 11 shows the GVSM in a) against the matching HydraProbe reading and in b) against the mean of the four readings collected at a sampling location.

From a comparison of Fig. 11 with Fig. 10, it is evident that the agreement between the HydraProbe readings and GVSM is equal or better than the results obtained for the 2009 and 2015 ThetaProbe data. Also noticeable in Figs. 10 – 11 are the little





differences among the distributions of the data points belonging to individual stations, which again may question the added value of station-specific calibration functions. However, because of the larger number of GVSM - HydraProbe pairs (> 86) and larger soil moisture range for individual stations, we decided to develop for the HydraProbe measurements station-specific calibration functions. Users of the dataset have the choice to apply the calibration function that suits their application best.

The development of calibration functions for the ThetaProbe and HydraProbe measurements consists of fitting linear regression coefficients ($a$ and $b$) following the same procedure as described in section 3.3 for the 5TM measurements. Table 4 provides the $\mu$ and $\sigma$ of the coefficients for the ThetaProbe functions along with performance metrics. Table 5 lists the same information for the HydraProbe and in Fig. 12 the probe measurements calibrated with the field campaign-specific function are plotted against the GVSM.

The performance metrics presented in Tables 4 and 5 show that the matching probe ('site') and GVSM measurements generally led to better performance except for the 2009 field campaign, for which possible explanations are given above. Of the field campaign calibrations, the calibration developed for the HydraProbe (2016-2017) led to the best results with a RMSE of 0.032 $m^3$ $m^{-3}$ versus 0.041 $m^3$ $m^{-3}$ for 2015 and 0.048 $m^3$ $m^{-3}$ for 2009. A very good match of the HydraProbe with the GVSM is obtained for ITC_SM10 with a RMSE of 0.022 $m^3$ $m^{-3}$. The explanation could be a combination of sandy soil and yearly cultivated land, which reduces disturbances due to soil clod and plant root, and is favourable for reliable soil sampling. Under more difficult circumstances, such as the loamier soil with clods at ITC_SM07, the metrics are closer to but still better than the ones obtained for the 2009 and 2015 field campaigns.

## 5. Quality assurance procedures

### 5.1 Maintenance monitoring stations

The monitoring stations were visited for maintenance operations and retrieving the recorded data twice a year from the inception of the network till 2011. This reduced to yearly visits up to 2014 and returned back to visits at least twice a year from 2015 till now. Visits are planned at the start (April/May) and at the end (October/November) of the growing season. The standard EM50 loggers are not connected to a telecommunication network and the recorded data is retrieved on-site using a laptop equipped with the ECH2O Utility software (METER Group, 2019). The internal memory of the EM50 loggers is sufficient for 12.8 months of operations in the default setup with five probes recording every 15 minutes, before the oldest data starts to be overwritten.

Each site visit includes the following standard activities: i) retrieving the recorded data, ii) making preliminary checks of the data quality, iii) taking photographs of the measurement setup and its surroundings, iv) replacing silica gel bags as desiccant (in recent years) and v) taking notes of the undertaken maintenance operations and any specifics related to the data quality or the measurement setup. Typical maintenance practices include replacing batteries, sensors and loggers, reconnecting probes, remounting loggers, and drying and cleaning loggers. Dates on which major changes were made to the measurement setups,





e.g. installation of stations, relocation of stations within the same field or to a different field, replacement of sensors, are specified in Table S2 and reported as a data quality (DQ) flag in the dataset (discussed below).

### 5.2 Data processing and flagging


The datasets are made available at three processing levels referred to as raw, processed and calibrated data. The raw data from the monitoring stations are the native EM50 data logger files organized per monitoring station. These files are in the MS Excel 97-2003 format and have two worksheets, of which one includes the unprocessed data (digital numbers) and the other holds soil moisture and soil temperature measurements converted from the digital numbers using default calibration functions. For
details, we refer to the EC-TM and 5TM (METER Group, 2019) manuals (Decagon Devices, 2008 and 2017) and the readme provided together with the dataset. The processed data is developed from the raw soil moisture and temperature data and checked for missing time stamps, missing values are replaced with -99.999, time stamps are converted to a consistent format [dd-mm-yyyy hh:mm] and placed in a chronological order starting with January 1 of the year the station was installed till December 31 of the year operations were stopped or 2020. The resulting data files, one for each station, are converted into
CSV files with suffix _pd. The calibrated data is obtained through the application of the developed calibration functions (section 3.3) to the processed data and is included in the CSV files with suffix _cd.

DQ flags are created, providing details related to the measurement setup and the reliability of the calibrated data in an automated manner. The DQ flags are documented in separate CSV files with suffix _fg. The files include 4 sets of flags indicative for the quality of the i) soil moisture and ii) soil temperature data, iii) particularities related to the measurements
setup and vi) probe type. The DQ flags start respectively with 'SM', 'ST', 'MS' and 'PR', followed by 5 integers each referring to one of the respective 5 ports of EM50 data logger. The automated quality control procedure reported in Dorigo et al. (2013, 2021) is largely adopted for i) and ii), whereby the flags requiring external data are omitted. Table 6 lists the flags and their descriptions for the four respective flag types. Only the highest digit is visible within the dataset, implicating that the order of the flags associates with an increase in concern for the data quality.

In the case of the field campaign data, the raw data is organized on a yearly basis. The processed data consists of soil moisture contents obtained through application of the default calibration function to the native probe readings and the calibrated data are the processed soil moisture contents to which field campaign-specific calibration functions have been applied. Details on the data processing can be found in readme document accompanying the dataset. Both the processed and calibrated data are combined in a single comma separated values (CVS) formatted file with suffix _pd_cd for the stations where field campaigns
took place.

### 5.3 Known issues

Long-term operation of in-situ monitoring networks goes hand in hand with measurement uncertainties. In this section, we would like to make data users aware of issues specific for the Twente dataset, which can be separated into items related to the instrumentation and to the measurement setup.



EC-TM and 5TM probes have been used in the monitoring network, and other than the points earlier described in Benninga et al. (2018), these probes are calibrated at the factory using standards among which one with an $\varepsilon_r$ of 40 as the highest. This means that native EC-TM and 5TM probe readings above 0.587 m³ m⁻³ and 0.510 m³ m⁻³, respectively, reach beyond the calibration domain of the sensors.

Another point of attention has been the inconsistency in the firmware of probes produced in 2013 with the latest version 4.0

and the earlier ones, of which the latter have not been deployed in the Twente network. In 2013, the manufacturer modified their calibration process to include two dielectric standards that turned out to overestimate the $\varepsilon_r$ between 10 and 20 (Decagon costumer notification 2014). We have applied the function supplied by the manufacturer to convert the 5TM readings and developed calibration functions for both probe versions. The 'all soils' calibration coefficients for firmware v4.0 are listed in Table 2 and applied accordingly.

Specific for the measurement setup of the Twente monitoring network is the placement of the instrumentation at the border of parcels, which inevitably has consequences for the representativeness for the field. Large differences in the meteorological inputs, e.g. precipitation and incoming solar radiation, are not expected, but small scale topography, spatially variable soil texture, differences in land cover and the local drainage infrastructure may cause discrepancies between the VSM at the border and inside of the field. The field campaigns described in Section 4 have been conducted to address this issue. Benninga et al.

(2020) have shown that comparisons of the 2016 and 2017 field campaign data against the measurements collected at stations yield RMSEs varying from 0.037 m³ m⁻³ to 0.068 m³ m⁻³ and $R^2$s of 0.56 up to 0.81. These levels of uncertainty are larger but yet of a similar magnitude as the performance metrics reported for the probe calibrations in Sections 3.3 and 4.3.

## 5. Time series

Figure 13 shows the soil moisture measured at depths of 5 cm, 10 cm, 20 cm, 40 cm and 80 cm over the period from January

2016 till June 2020 for monitoring stations ITC_SM10 (Fig. 13b), ITC_SM14 (Fig. 13c) and ITC_SM17 (Fig. 13d). The groundwater level measured at the DINOLoket well closest to the respective soil moisture monitoring station (see supplement Table S1) is shown in the same plots and the upper panel presents the daily precipitation and daily air temperature as averages of the measurements collected at the three KNMI automated weather stations in the region.

Overall the time series confirm the seasonal dynamics of wet soils and high groundwater levels in winters, and dry

circumstances with low groundwater levels during summers. Also expected is the stronger response to precipitation of the soil moisture contents measured closest to the surface, whereas at 80 cm mainly seasonal variations are noted. Specifically in the 80 cm soil moisture content the effect of the 2018, 2019 and 2020 droughts is visible, while the top soil (5 and 10 cm) dries out during the summer period virtually every year.

At the same time, substantial spatial differences can be noted between the three monitoring stations, which are situated 25 km

– 30 km apart at elevations of 10 m to 15 m a.m.s.l.. For instance, in Fig. 13c (ITC_SM14) the 80 cm soil moisture content remained at a high level even during the peak of the 2018 drought, whereas deep drops are observed in Figs. 13b (ITC_SM10)

and 13d (ITC_SM17). These measurements demonstrate that the position within a catchment is an important factor for the impact drought has locally, even though drought may be seen as a regional-scale process.

Somewhat surprising in the plots is the response of the groundwater level to precipitation. In all three groundwater

measurement series increments can be identified after precipitation events, whereas the soil moisture at 80 cm primarily displays seasonal variations and individual events are hardly noticeable. To take this a step further and explore the relationship with soil moisture, Table 7 presents the $R^2$ values computed between the measurements at specific depths and groundwater levels. Indeed, the $R^2$ values support the above observation. The shallower 40 cm soil moisture content yields the highest $R^2$ and not the deeper 80 cm measurements. This is in spite of the fact that the highest groundwater levels remain substantially

below the surface with -112 cm, -83 cm and -71 cm recorded as the highest levels near ITC_SM10 (Fig 13b) ITC_SM14 (Fig. 13c) and ITC_SM17 (Fig. 13d), respectively.

Another interesting feature is that the soil moisture at 5 cm and 10 cm are still reasonably correlated with the groundwater levels. Hence, there is a research opportunity to further investigate the potential of the near-surface soil moisture, observable from space, for supplying information on the groundwater level. Similar work has previously been conducted by Sutanudjaja

et al. (2013), who estimated groundwater level across the Rhine-Meuse river basin using time series of soil water index retrieved from coarse resolution scatterometer data. The present dataset allows for more detailed investigations of the relationship between phreatic groundwater and profile soil moisture. Moreover, the spatial measurement density of the Twente network, the access to the other relevant data documented in this manuscript and the availability of higher resolution soil moisture products (e.g. Bauer-Marschallinger et al., 2019, Das et al. 2019) makes it possible to extend to sub-catchment scale

applications.

## 6. Data availability

The raw , processed , and calibrated soil moisture and temperature station data from 2008 till 2020, the DQ flags, the photos taken during field visits as well as the intensive measurements collected during the 2009, 2015, 2016 and 2017 field campaigns are publicly available at https://doi.org/10.17026/dans-xr2-m6d8 (Van der Velde and Benninga, 2022). Folder and file

structures as well as the processing steps are described in a readme file. The locations of the measurement are given in geographic (WGS84) and map projected coordinates (Amersfoort/RD new). Table 8 lists the third-party datasets that are available for the study region, which may support use of the published soil moisture and temperature datasets.

## 7. Summary

Soil moisture and temperature profile measurements from 2008 till 2020 have been automatically collected at 15 minute

intervals by a network of 20 monitoring stations spread across the Twente region and neighbouring regions in the east of the Netherlands. The monitoring stations are mostly placed at the border of privately owned parcels used for agriculture with, in

order of occurrence, grass, maize, cereals, potato and natural vegetation as land covers. The experimental setup includes METER Group (formerly: Decagon) EC-TM and its successor 5TM capacitance probes installed at soil depths of 5 cm, 10 cm, 20 cm, 40 cm and 80 cm. Soil-specific calibration functions have been developed under controlled laboratory conditions for both probe types suggesting accuracies of 0.023 $m^3$ $m^{-3}$ and 0.028 $m^3$ $m^{-3}$ for the EC-TM and 5TM, respectively. Quality-controlled and calibrated datasets as well as the native data records, and field photos are made available.

In addition, field campaign data covering the growing seasons of 2009, 2015, 2016 and 2017 are described and disclosed, during which soil moisture content was measured with handheld probes (Delta-T ThetaProbe, Type ML2, and Stevens HydraProbe) and a gravimetric method on a total of 28 fields near twelve different monitoring stations. Pairs of gravimetrically determined soil moisture and probe readings were used to establish calibration functions for both the ThetaProbe and HydraProbe. The accuracies obtained for the probe calibrations varied from 0.048 $m^3$ $m^{-3}$ for the ThetaProbe measurements in 2009 up to 0.032 $m^3$ $m^{-3}$ for the HydraProbe measurements collected in 2016-2017.

Further, descriptions of open thirty-party datasets are provided to support the use of the Twente soil moisture and temperature measurements beyond the scope for which the network was originally established: the validation of coarse resolution satellite data products. Scientists and professionals worldwide are invited to make free use of the datasets disclosed with this contribution. We welcome any comments or suggestions that can help improve the quality and usability of the datasets. The data collection with the Twente network continues, but plans are underway to update the design of the network to the contemporary societal and scientific needs. This may include flood and drought analyses, and high resolution satellite product validation. We invite scientists and professionals worldwide to make free use of the datasets collected in the framework of the Twente monitoring network for any purpose it may fit under a Creative Commons, CC BY 4.0 license.

**Author contribution**

RV and HJB contributed to the fieldwork, data processing, data quality control, conceptualization and writing of the paper. HJB led the data quality control, and RV coordinated and led the writing of the paper.

**Competing interests**

The authors declare that they have no conflict of interest.

**Acknowledgements**

The authors thank the farmers who provided free access to the parcels where the monitoring stations have installed. The Royal Netherlands Academy of Arts and Sciences (KNAW) is acknowledged for the support via the small data project (Klein Data Project) programme for making the dataset available through its DANS (Data Archiving and Networked Services) platform, project number KDP002. Laura Dente, Zoltan Vekerdy, and Bob Su are acknowledged for their role in



the development and involvement of the monitoring network till 2012. Murat Ucer is mentioned for his contribution to the field data collection. Further, the authors would like to thank all the students and researchers who participated in the field data collection over the years.

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





**List of tables**

**Table 1: Soil, land use and maintenance characteristics of the stations. The soil descriptions are from BOFEK2020 (Heinen et al., 2021). The land uses for 2009 – 2020 apply to adjacent fields and are from the crop parcel registry (Ministry of Economic Affairs and Climate Policy, 2021). Table classification: green stands for grass, orange stands for maize, red stands for potato, yellow stands**
**for cereal, purple stands for other crops, brown stands for forest. Relocations of stations are noted by letters, which correspond to the locations in Fig. 2. Other maintenance practices are noted by asterisks and are specified in supplement Table S2.**

**Table 2: Mean ($\mu$) regression coefficients and their standard deviations ($\sigma$) fitted through pairs of GVSM and 5TM VSM measured in the laboratory on soil collected at sites ITC_SM03, ITC_SM07 and ITC_SM08. Performance metrics, RMSE, ME and $R^2$, follow from the validation. $n$ stands for the number of matchups.**

**Table 3: Overview of the soil moisture field campaigns conducted at fields adjacent to monitoring stations. In the far right column, the number in parenthesis stands for the number of sampled fields and the letter represents the land cover at the start of the campaign (g = grassland, m = maize, f = forest, fw = fallow winter wheat, w = winter wheat, p = potato).**

**Table 4: Mean ($\mu$) and standard deviation ($\sigma$) of regression coefficients obtained for pairs of GVSM and ThetaProbe VSM and associated performance metrics (RMSE, ME, $R^2$) for measurements taken during the 2009 and 2015 field campaigns. Two matching**
**ThetaProbe values are used: i) a reading next to the soil sample (in the table: site), ii) the mean of all readings taken at the sampling point (in the table: mean). $n$ stands for the number of matchups.**

**Table 5: Similar to Table 4, but for calibrations of 2016-2017 HydraProbe measurements. In this case, calibration functions were also developed for individual stations.**

**Table 6: Soil moisture and temperature data quality, measurements setup and probe type flags included in the DQ files created**
**based on the calibrated data files.**

**Table 7: $R^2$ computed between soil moisture measured at specific depths and groundwater level at the well nearest to the soil moisture monitoring station available in DINOloket (see Supplement Table S1). The time series are shown in Figure 13. $n$ stands for the number of groundwater and soil moisture data pairs.**

**Table 8: Open third-party datasets available for the study region described in Section 2.**






**Table 1: Soil, land use and maintenance characteristics of the stations. The soil descriptions are from BOFEK2020 (Heinen et al., 2021). The land uses for 2009 – 2020 apply to adjacent fields and are from the crop parcel registry (Ministry of Economic Affairs and Climate Policy, 2021). Table classification: green stands for grass, orange stands for maize, red stands for potato, yellow stands for cereal, purple stands for other crops, brown stands for forest. Relocations of stations are noted by letters, which correspond to the locations in Fig. 2. Other maintenance practices are noted by asterisks and are specified in supplement Table S2.**

| Station | Soil type class | Soil description translated from Dutch (BOFEK2020 classification code) | 2009 | 2010 | 2011 | 2012 | 2013 | 2014 | 2015 | 2016 | 2017 | 2018 | 2019 | 2020 |
|---|---|---|---|---|---|---|---|---|---|---|---|---|---|---|
| ITC_SM01 | Sandy | Highly loamy sandy soil with clay cover (3002) | a | | | | | * | b | | | | | |
| ITC_SM02 | Sandy | Highly loamy soil with man-made thick earth (3005) | a | | | | b | | * | * | * * | | | |
| ITC_SM03 | Sandy | Highly loamy sand with clay cover (3002) | a | | | | | * | | | b | | | |
| ITC_SM04 | Loamy | Tertiary clay (5003) | a | | | | | | | b * | | | * | |
| ITC_SM05 | Sandy | Highly loamy soil with man-made thick earth (3005) | a | | | | | | | * | b | c | | |
| ITC_SM06 | Partly organic | Sandy cover on partly organic soil (2001) | a | | | | * | | * | b | | | | |
| ITC_SM07 | Sandy | Highly loamy sand with clay cover (3002) | a | | | | | * | b | | | | | |
| ITC_SM08 | Sandy | Weakly loamy sand (3015) | | * | | | | | | * | | | | |
| ITC_SM09 | Sandy | Weakly loamy soil with man-made thick earth (3012) | | | | | * * | | * | | | * | | |
| ITC_SM10 | Sandy | a & b: Highly loamy sand (3004) c: Highly loamy sand (3021) | a | | | | * | b | | c * | | | | |
| ITC_SM11 | Sandy | a & b: Weakly loamy soil with man-made thick earth (3012) c: Highly loamy sand (3004) | a | | | * | | | b* | c * | | | | |
| ITC_SM12 | Clayey | Clay on sand (4022) | | | | | * | | * | * | | | | |
| ITC_SM13 | Sandy | Weakly loamy sand (3015) | | | | | | * | | * | * | | | |
| ITC_SM14 | Sandy | Highly loamy sand (3021) | a | * | b | | * | * | | c | * | | | |
| ITC_SM15 | Sandy | Highly loamy sand with clay cover (3002) | a | | | | | | | b* | | | * | |





| ITC_SM16 | Partly organic | Sandy cover on partly organic soil (2001) | | | * | * | | * | | | * | * | | |
|---|---|---|---|---|---|---|---|---|---|---|---|---|---|---|
| ITC_SM17 | Sandy | Weakly loamy sand (3015) | a | | b | | | | * | c | | | | |
| ITC_SM18 | Sandy | Highly loamy sand (3021) | | | | | | * | * | | | | | | * |
| | | | | | | | | | | | | | | |
| | | | | | | | | | | | | | | |
| ITC_SM19 | Sandy | Highly loamy sand (3004) | | | | | | * | | | | | | | |
| | | | | | | | | | | | | | | |
| | | | | | | | | | | | | | | |
| ITC_SM20 | Sandy | Coarse sandy sand (3003) | | | | | | * | | | | | | | |
| Hupsel | Sandy | Highly loamy sand (3004) | | | | | | | | | | | | | |
| | | | | | | | | | | | | | | |
| Twenthe airport | Sandy | Weakly loamy sand (3014) | | | | | | | | | | | * | * | |


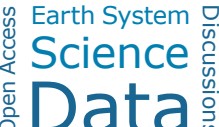

**Table 2: Mean ($\mu$) regression coefficients and their standard deviations ($\sigma$) fitted through pairs of GVSM and 5TM VSM measured in the laboratory on soil collected at sites ITC_SM03, ITC_SM07 and ITC_SM08. Performance metrics - RMSE, ME and R$^2$ - follow from the validation. $n$ stands for the number of matchups.**

| | | $a$ ($m^3\ m^{-3}$) | | $b$ (-) | | RMSE | ME | $R^2$ |
|---|---|---|---|---|---|---|---|---|
| Set | n | $\mu$ | $\sigma$ | $\mu$ | $\sigma$ | ($m^3\ m^{-3}$) | ($m^3\ m^{-3}$) | - |
| ITC_SM03 | 38 | 0.00423 | 0.00186 | 1.87 | 0.0165 | 0.0237 | 0.000 | 0.927 |
| ITC_SM07 | 32 | 0.0214 | 0.00307 | 1.77 | 0.0208 | 0.0303 | 0.000 | 0.883 |
| ITC_SM08 | 29 | 0.0546 | 0.00510 | 1.52 | 0.0369 | 0.0315 | 0.000 | 0.786 |
| All soils[*] | 99 | 0.0200 (-0.0217) | 0.000958 | 1.76 (1.63) | 0.00737 | 0.0277 | 0.000 | 0.884 |

[*] In parenthesis are the calibration coefficients for the 5TM probes with firmware v4.0.






**Table 3: Overview of the soil moisture field campaigns conducted at fields adjacent to monitoring stations. In the far right column, the number in parenthesis stands for the number of sampled fields and the letter represents the land cover at the start of the campaign (g = grassland, m = maize, f = forest, fw = fallow winter wheat, w = winter wheat, p = potato).**

| Year | Period | Days | Probe | Stations (fields) |
|---|---|---|---|---|
| 2009 | 22 Sept – 28 Oct | 5 | ThetaProbe | ITC_SM03 (1g), 05 (2g/1m), 07 (3m), 08 (1g/1m/1f), 11 (2g/1f), 12 (3g), 17 (3g), 18 (3g) |
| 2015 | 11 Sept – 3 Nov | 11 | ThetaProbe | ITC_SM03 (1g), 04 (2g), 05 (3g), 07 (3m), 08 (1m/1g), 09 (fw2) |
| 2016 | 25 May– 11 Nov | 15 | HydraProbe & ThetaProbe | ITC_SM02 (1g/1m), 07 (2m), 10 (1m/1p) |
| 2017/2018[*] | 7 April – 16 Nov | 14 | HydraProbe | ITC_SM02 (1g/1m), 03 (1g), 07 (2m), 10 (2m) |

[*] In 2018 a limited number of fields were sampled on February 2nd and April 10th.






**Table 4: Mean (μ) and standard deviation (σ) of regression coefficients obtained for pairs of GVSM and ThetaProbe VSM and associated performance metrics (RMSE, ME, R². ) for measurements taken during the 2009 and 2015 field campaigns. Two matching ThetaProbe values are used: i) a reading next to the soil sample (in the table: site), ii) the mean of all readings taken at the sampling point (in the table: mean). *n* stands for the number of matchups.**

| | | | $a$ (m³ m⁻³) | | $b$ (-) | | RMSE | ME | R² |
|---|---|---|---|---|---|---|---|---|---|
| Set | *n* | *Matchup* | μ | σ | μ | σ | m³ m⁻³ | m³ m⁻³ | - |
| 2009 | 93 | site | 0.0686 | 0.00139 | 0.920 | 0.00532 | 0.0522 | -0.001 | 0.732 |
| | | mean | 0.0498 | 0.00130 | 0.992 | 0.00484 | 0.0477 | -0.001 | 0.780 |
| 2015 | 166 | site | -0.0128 | 0.000735 | 1.09 | 0.00267 | 0.0411 | 0.000 | 0.875 |
| | | mean | -0.00899 | 0.000733 | 1.09 | 0.00277 | 0.0417 | 0.000 | 0.871 |






**Table 5: Similar to Table 4, but for calibrations of 2016-2017 HydraProbe measurements. In this case, calibration functions were also developed for individual stations.**

| Set | n | Matchup | $a$ (m³ m⁻³) | | $b$ (-) | | RMSE | ME | $R^2$ |
| | | | $\mu$ | $\sigma$ | $\mu$ | $\sigma$ | (m³ m⁻³) | (m³ m⁻³) | - |
|---|---|---|---|---|---|---|---|---|---|
| ITC_SM02 | 92 | site | 0.0738 | 0.000980 | 0.849 | 0.00670 | 0.0324 | 0.000 | 0.877 |
| | | mean | 0.0550 | 0.000546 | 0.947 | 0.00352 | 0.0289 | 0.000 | 0.897 |
| ITC_SM03 | 12 | site | 0.0875 | 0.00527 | 0.780 | 0.0196 | 0.0378 | 0.002 | 0.903 |
| | | mean | 0.0923 | 0.00833 | 0.836 | 0.0405 | 0.0425 | 0.003 | 0.903 |
| ITC_SM07 | 86 | site | 0.0797 | 0.00214 | 0.788 | 0.00988 | 0.0384 | 0.000 | 0.805 |
| | | mean | 0.0865 | 0.00203 | 0.801 | 0.00956 | 0.0421 | 0.000 | 0.759 |
| ITC_SM10 | 92 | site | 0.0420 | 0.000427 | 0.961 | 0.00388 | 0.0217 | 0.000 | 0.929 |
| | | mean | 0.0621 | 0.000620 | 0.927 | 0.00453 | 0.0329 | 0.000 | 0.833 |
| 2016-2017* all | 285 | site | 0.0637 | 0.000319 | 0.860 | 0.00196 | 0.0323 | 0.000 | 0.881 |
| | | mean | 0.0669 | 0.000311 | 0.890 | 0.00187 | 0.0351 | 0.000 | 0.858 |

* Three pairs collected on fields adjacent to ITC_SM05 were included in the regional calibration (2016-2017).






**Table 6: Soil moisture and temperature data quality, measurements setup and probe type flags included in the DQ files created based on the calibrated data files.**

| Flag type | Flag | Method | Description |
|---|---|---|---|
| Soil moisture (SM) and temperature (ST) data quality | 0 | n/a | Normal operations |
| | 1 | Range verification | Soil moisture below 0.0 $m^3m^{-3}$ or soil temperature below -20ºC |
| | 2 | Range verification | Soil moisture above 0.7 $m^3m^{-3}$ or soil temperature above 50ºC |
| | 3 | Spectrum based | Spike detected |
| | 4 | Spectrum based | Negative break (drop) |
| | 5 | Spectrum based | Positive break (jump) |
| | 6 | Spectrum based | Constant low values following a negative break |
| | 7 | Spectrum based | Saturated plateau following a positive break |
| | 9 | n/a | No data |

| Flag type | Flag | Description |
|---|---|---|
| Measurement setup (MS) | 0 | Normal |
| | 1 | Installation of the station |
| | 2 | Replacement of the sensor |
| | 3 | Relocation within the same field |
| | 4 | Relocation to a different field |
| | 9 | No measurements |

| Flag type | Flag | Description |
|---|---|---|
| Probe type (PR) | 0 | No probe |
| | 1 | EC-TM |
| | 2 | 5TM firmware version 2013 |
| | 3 | 5TM firmware v4.0 |



**Table 7: R$^2$ computed between soil moisture measured at specific depths and groundwater level at the well nearest to the soil moisture monitoring station available in DINOloket (see Supplement Table S1). The time series are shown in Figure 13. *n* stands for the number of groundwater and soil moisture data pairs.**

| | | Soil moisture measured at depth | | | | |
|---|---|---|---|---|---|---|
| *Station* | *n* | *5 cm* | *10 cm* | *20 cm* | *40 cm* | *80 cm* |
| *ITC_SM10* | 1490 | 0.515 | 0.499 | 0.714 | **0.779** | 0.758 |
| ITC_SM14 | 1338 | 0.722 | 0.575[*] | 0.709 | **0.782** | 0.527 |
| ITC_SM17 | 1332 | 0.405 | 0.509 | 0.628 | **0.853** | 0.851 |

[*] obtained for 1251 pairs.



**Table 8: Open third-party datasets available for the study region described in Section 2.**

| Name | Responsible institute(s) | Data address and instructions | Available formats |
|---|---|---|---|
| Actueel Hoogtebestand Nederland | RWAs, provinces, Directorate-General for Public Works and Water Management | https://www.pdok.nl/introductie/-/article/actueel-hoogtebestand-nederland-ahn3- Under the tab 'Downloads' individual tiles can be obtained and under 'Geo Services' links to the entire dataset are provided. | GeoTIFF WMS WFS WMTS WCS |
| BOFEK (Heinen et al., 2021) | Wageningen Environmental Research | https://www.wur.nl/nl/show/Bodemfysische-Eenhedenkaart-BOFEK2020.htm; The map and report can be found under downloads both for BOFEK2020 and BOFEK2012. | .gdb .shp |
| Land use file (Bestand Bodemgebruik) | Statistics Netherlands | https://www.pdok.nl/introductie/-/article/cbs-bestand-bodemgebruik; for the years 2010 and 2015 downloads as well as Geo Services are available. | .shp WMS WFS |
| Crop parcel registry (Basisregistratie Gewaspercelen) | Ministry of Economic Affairs and Climate Policy | https://data.overheid.nl/dataset/10674-basisregistratie-gewaspercelen--brp-; for the years 2009 – 2020 downloads are available at the tab 'Databronnen' and under 'INSPIRE Atom' and from 2016 also view services are available. | .gdb WMS WFS WMTS |
| DINOloket | Geological Survey of the Netherlands | https://www.dinoloket.nl/en; go to 'Subsurface data', apply a filter in the menu on the left and select one of the shapes in the menu on the right to order data for measurement locations. | .csv |
| Precipitation and weather data | Royal Netherlands Meteorological Institute | https://www.knmi.nl/nederland-nu/klimatologie-metingen-en-waarnemingen; for daily precipitation measurements select 'Dagwaarden neerslagstations' and for hourly weather data select 'Dagwaarden van weerstations'. | .txt |



**List of figures**



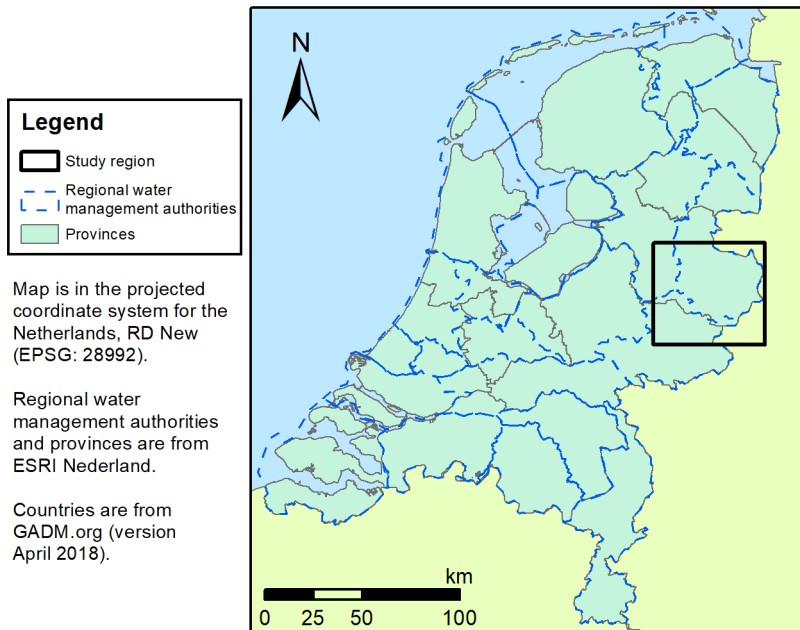

**Figure 1: Coverage of the monitoring network (study area) within the Netherlands and the boundaries of the RWAs and provinces.**

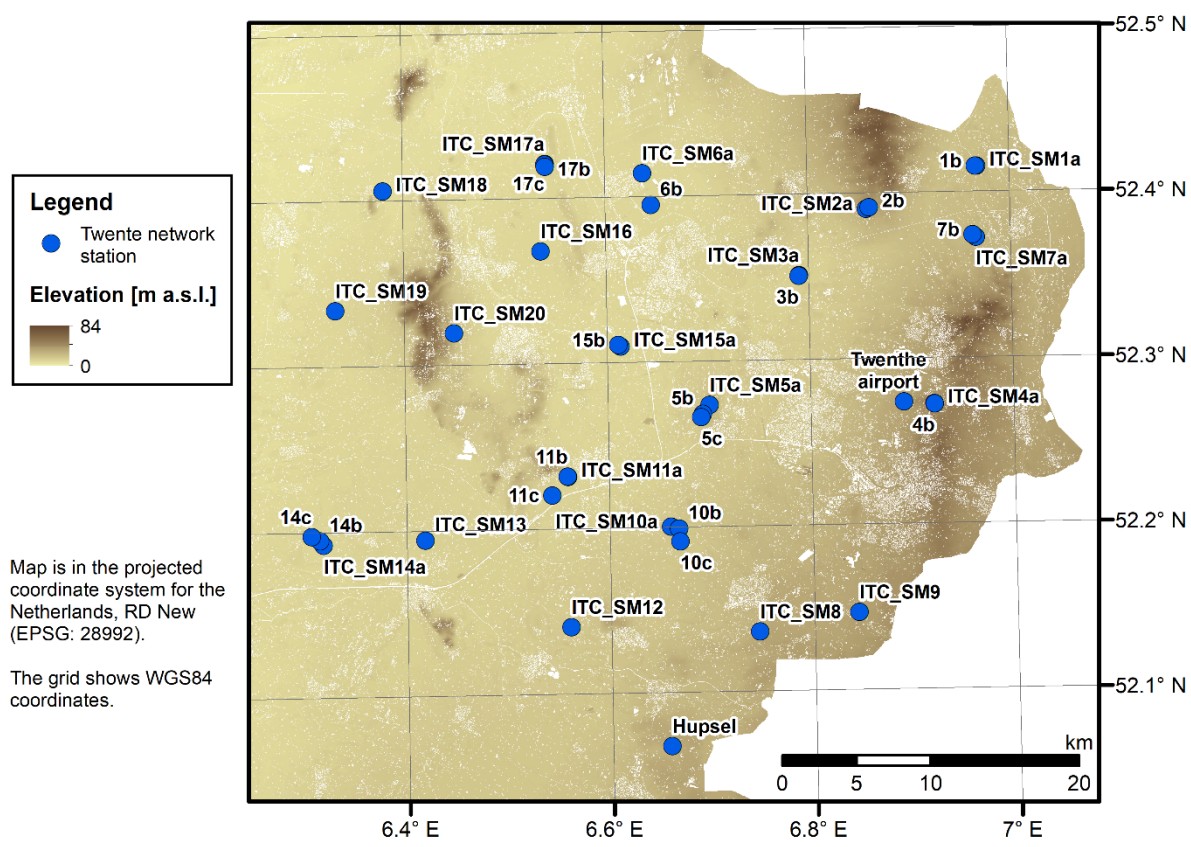

**Figure 2: The topography of the study area (source: 5 m spatial resolution AHN3; AHN, 2019) and the locations of the Twente soil moisture and temperature monitoring stations, whereby the number refers to the station ID and the letter to the specific location within the entire observation period.**

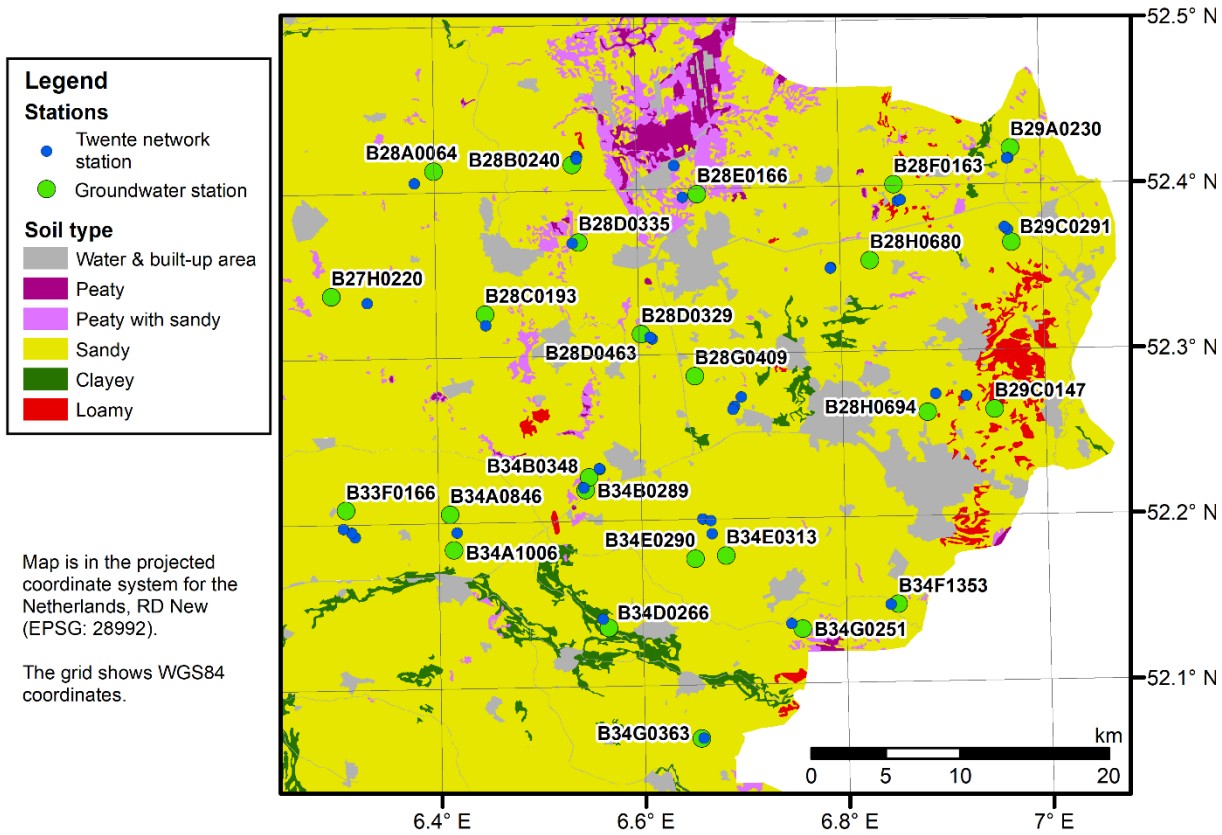

**Figure 3: Map with the major soil types in the study area (source: BOFEK2020; Heinen et al. 2021) and the groundwater monitoring wells near the Twente monitoring stations (source: DINOloket).**




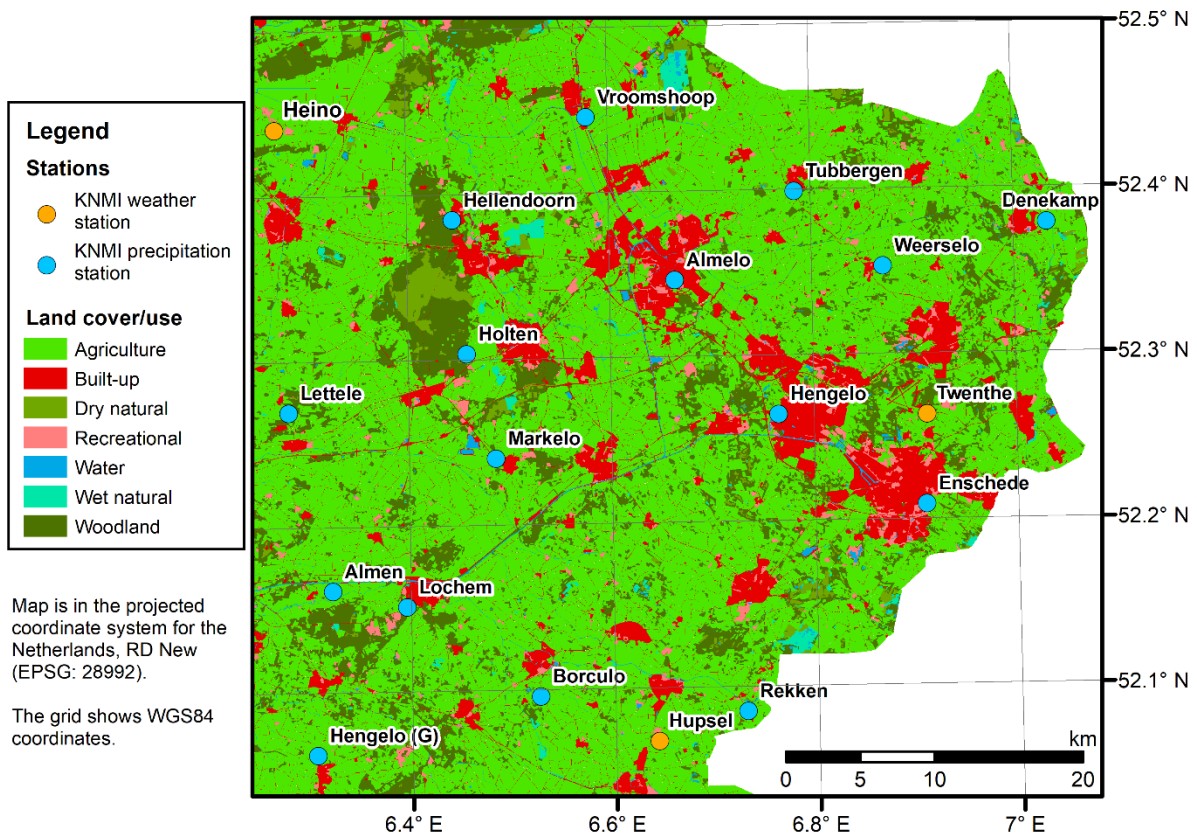

**Figure 4: Land use map of the study area and the location of KNMI meteorological measurement stations. Source: Statistics Nederlands and KNMI.**



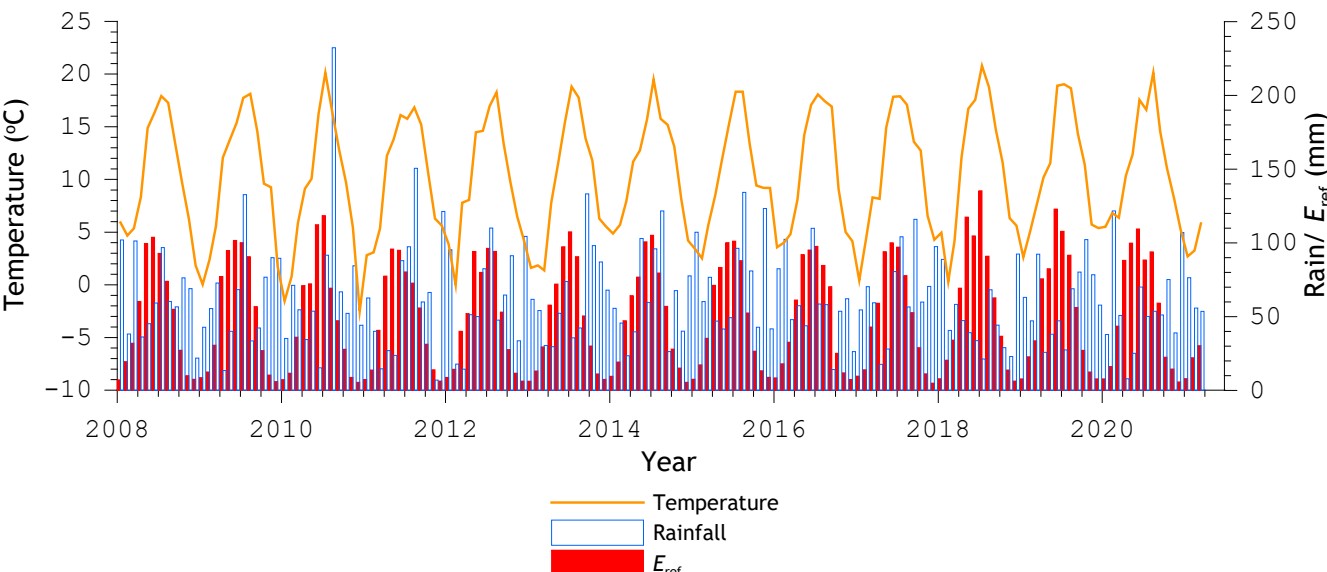

**Figure 5: Monthly mean 1.5 m air temperature, and monthly rainfall and $E_{\text{ref}}$ sums derived from the measurements collected at KNMI automated weather stations Heino, Hupsel and Twenthe.**




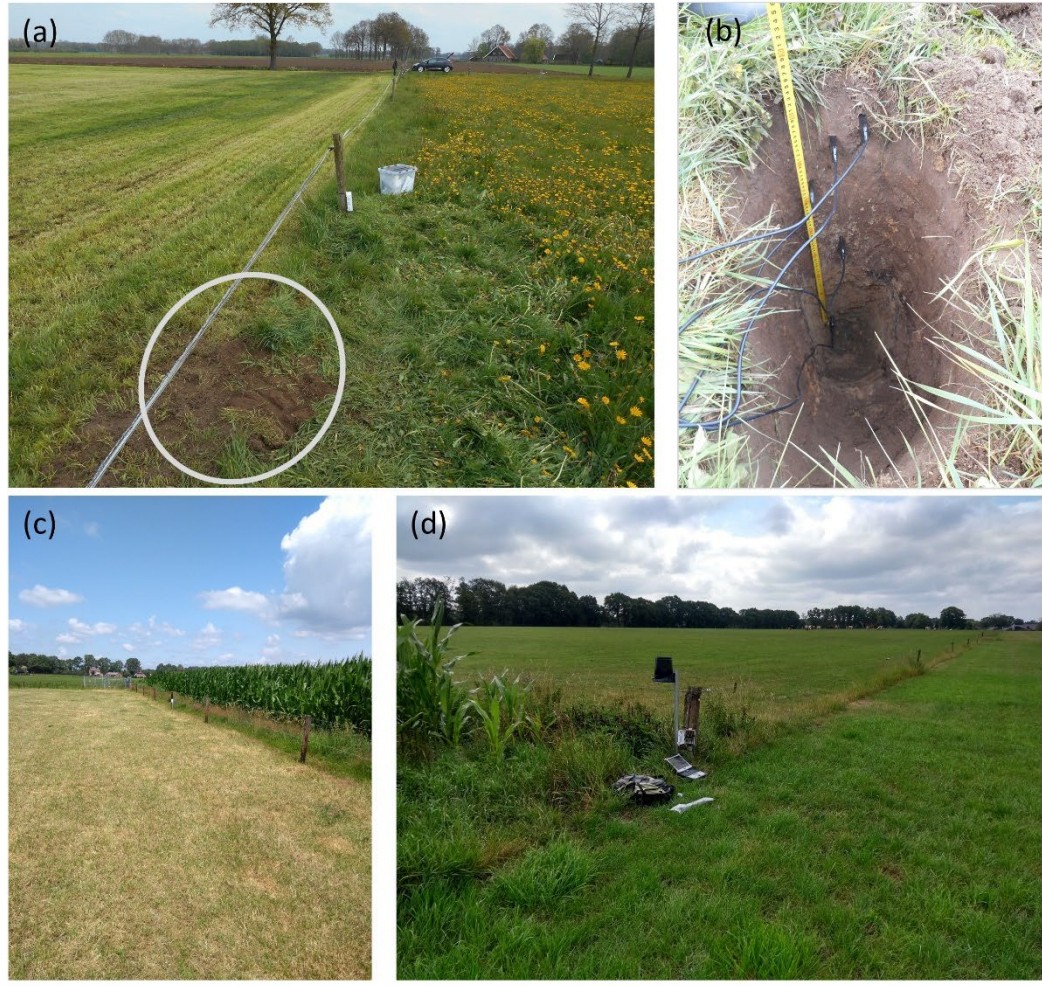

**Figure 6: Photos taken of (a and b) the reinstallation of ITC_SM03 on 2 May 2017, c) ITC_SM18 on 17 July 2019 and d) ITC_SM02 on 17 July 2019.**



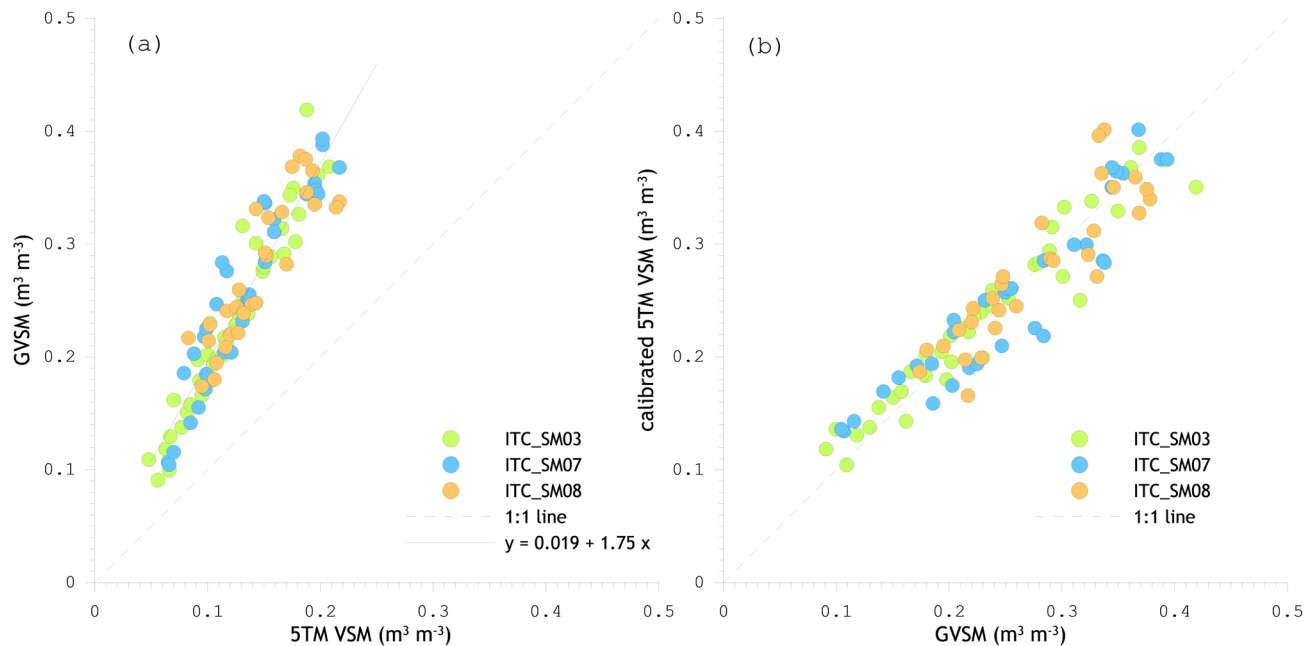


**Figure 7: a) Measurements of GVSM against 5TM VSM on soil collected at sites ITC_SM03, ITC_SM07 and ITC_SM08 and b) 5TM VSM with application of the 'all soils' calibration function against GVSM measurements.**

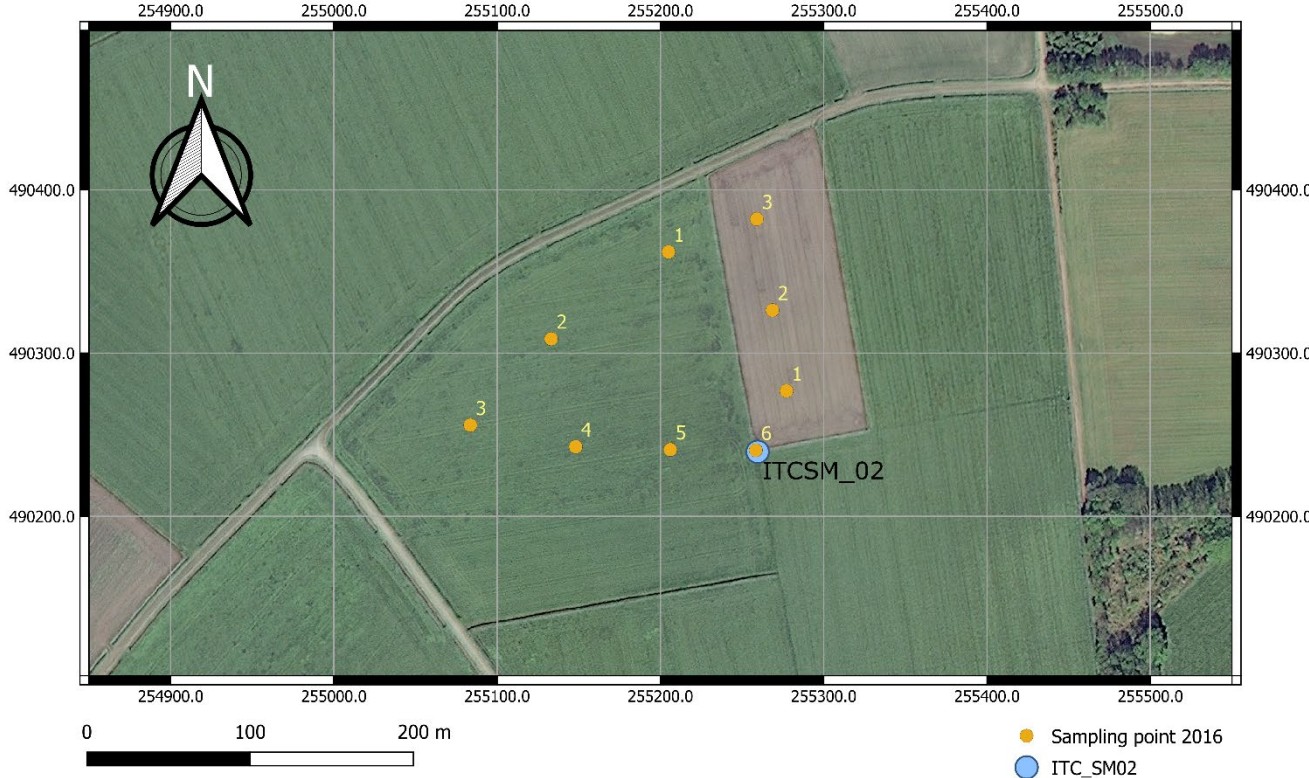

**Figure 8: Example (site: ITC_SM02) of the sampling strategy followed for the fields adjacent to permanent monitoring locations during field campaigns. Background is a 2.0 m resolution SuperView true colour composite of 25 April 2019 made available by Netherlands Space Office satellietdataportaal. Map is in the RD New (EPSG: 28992) projected coordinate system.**





**Figure 9: Schematization of the impedance probe and GVSM sampling strategy carried out at sampling points during the 2009, 2015, 2016 and 2017 field campaigns on grassland (a) and maize fields (b).**

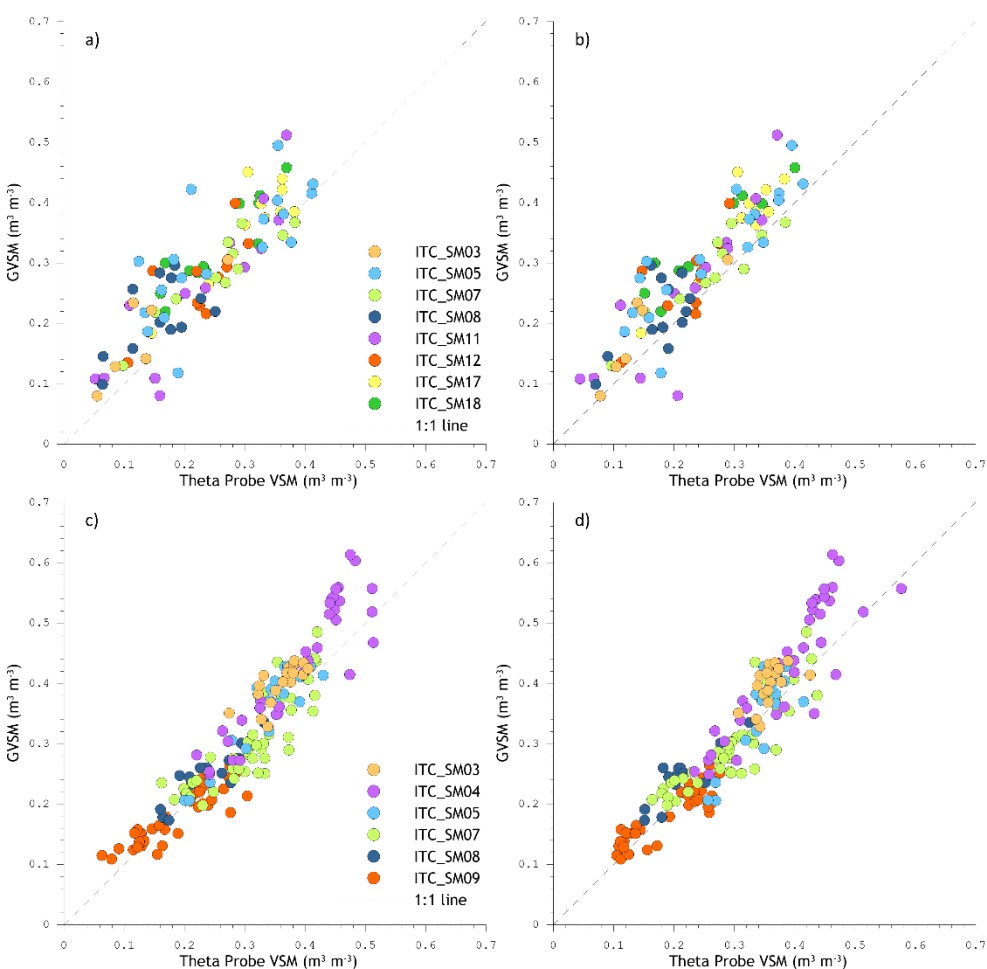

Figure 10: Scatter plots of the ThetaProbe VSM against GVSM collected during the 2009 (a and b) and 2015 (c and d) field campaigns. In subplots a) and c) are the ThetaProbe VSM reading taken next to a GVSM measurement. In subplots b) and d) are the mean of the ThetaProbe VSM readings taken at a sampling point.



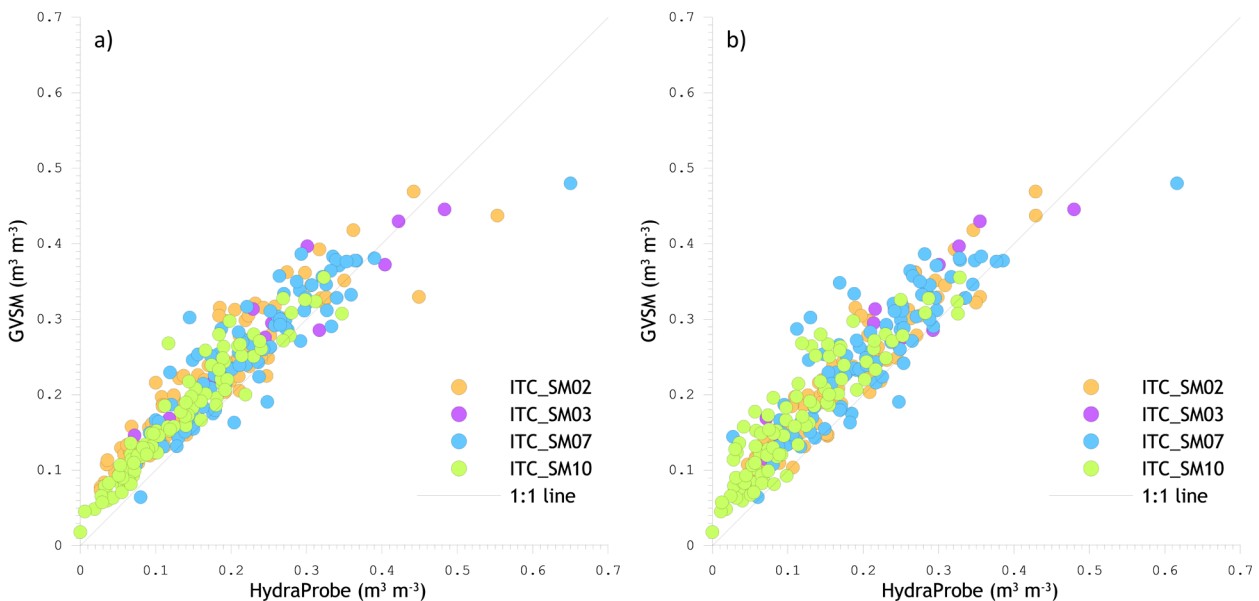

**Figure 11: Scatter plots with the HydraProbe VSM against the GVSM collected during the 2016 and 2017 field campaigns. In subplot a) is the HydraProbe VSM reading taken next to the GVSM measurement. In subplot b) is the mean of the HydraProbe VSM readings taken at the sampling point.**

.

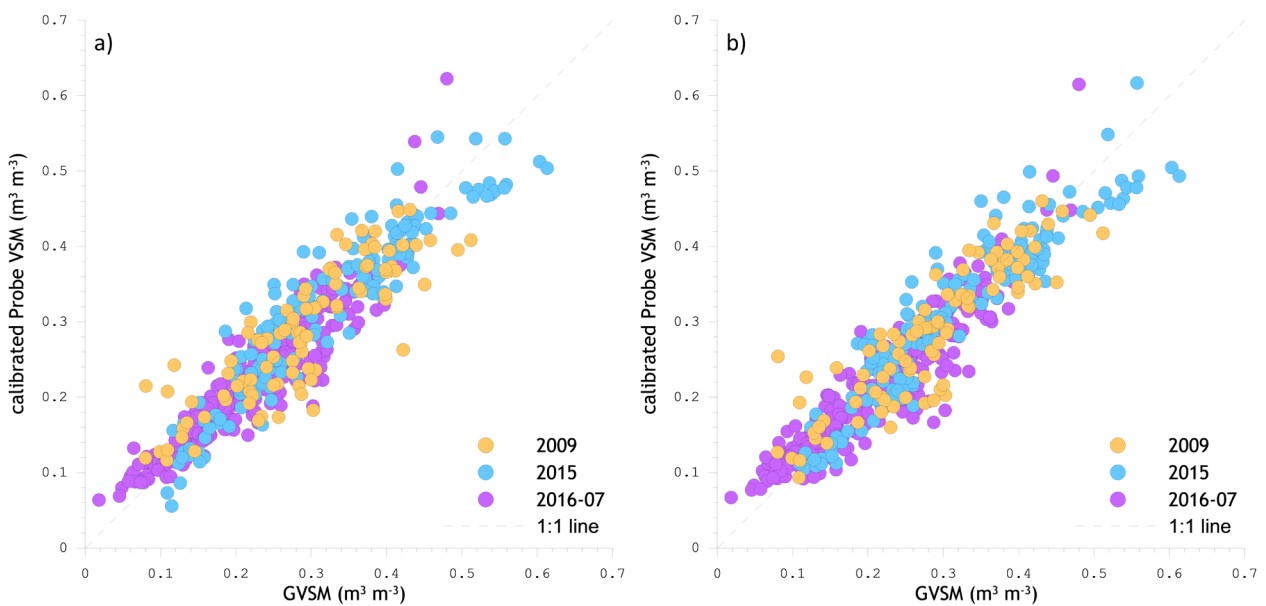

**Figure 12: Probe VSM with application of the field campaign-specific calibration functions plotted against GVSM. In subplot a) is the calibrated probe VSM derived from the reading taken next to the soil sample. In subplot b) is the calibrated probe VSM derived from the mean of the readings taken at a sampling point.**



**Figure 13: (a) Daily rainfall and daily air temperature as averages of the three KNMI automated weather stations. (b-d) Profile soil moisture measured at b) ITC_SM10, c) ITC_SM14, and d) ITC_SM17 and groundwater level measured in the nearest well available in DINOLoket (see supplement Table S1).**