# Peer review of "Twelve years profile soil moisture and temperature measurements in Twente, the Netherlands"

_Earth System Science Data, 2022_

## Referee Comment (RC1)

**Referee report on van der Velde et al. (2022), submitted to ESSD**

**Overall evaluation**

In their manuscript *"Twelve years profile soil moisture and temperature measurements in Twente, the Netherlands"*, R. van der Velde and colleagues present a corresponding dataset which is supplemented by several soil moisture measurement campaigns which add more points around the long-term measurements in terms of snapshots.

I congratulate the author team for a precious and useful dataset. The overall approach and design (a sparse network of profile probes) is not specifically innovative, yet it allows for interesting analyses and should be a useful asset for the scientific community. Most importantly, the author spent a lot of effort in the (re-)calibration of the soil moisture sensors by comparing the instrumental readings to gravimetric soil moisture measurements - an effort that is well-spent, and certainly nothing to be taken for granted.

The paper itself is mostly well-written, the accompanying dataset well-organised, complete, and contains both raw readings and processed data, conforming with various user requirements.

I still have some specific comments which you might classify as major, but which I am confident the authors will be able to consider in one way or the other. Most of them should be understood as suggestions to improve the manuscript.

**Specific comments**

**Structure and balance of the manuscript**

I think in the final parts of the manuscript, the structure should be adjusted to make it more straightforward. My suggestion is to drop section *"5.1 Maintenance monitoring stations"* (which is, in my opinion, not relevant in the context of this paper), and to combine section *"5.2 Data processing and flagging"* with the section *"Data availability"* (please note that the section numbering is wrong after section 5). Section *"5.3 Known issues"* deserves, in my view, a dedicated single subsection, but should be renamed to *"Data uncertainty"*. Please use this section to provide a more comprehensive synthesis on the two most important sources of uncertainty: the sensor calibration (most importantly the relationship between permittivity and volumetric soil moisture, but maybe also some words on the relationship between the primary measurement variable, typically raw voltage, and permittivity, which is assumed to be set by the manufacturer), and the spatial representativeness (see comment below). The section title *"Time series"* is not really adequate. Still, I think it is an important section as it presents an opportunity to highlight the scientific potential of your dataset (see comment "Underlying research questions and prospective application of the data").

Apart from the structure, parts of the paper could be more concise. Please reconsider whether some details on the instruments' construction design and the calibration procedure could be removed for the sake of readability.

**Underlying research questions and prospective application of the data**

In ll. 44-49, you describe previous research that was based on the presented dataset, and which already started more than 10 years ago. While this exemplifies potential use cases of the dataset, it also leaves the reader with a feeling that this dataset was only published after it had already been comprehensively used.

Of course, that is not true! All the more, however, it is important for this paper to highlight and to exemplify open research questions and application cases related to this data. I think the section that is now entitled "Time series" is a good opportunity to highlight interesting properties of your dataset - as you already started to do. Still, I am wondering why you only show 3 out of 20 stations and 4 out of 12 years. Of course, this paper should not be about a scientific analysis, and of course you should not just show time series of all stations at all depths for all years. But maybe you could demonstrate the potential of your dataset along some exemplary research questions, maybe also at different time scales? This is just a thought, but I think this section could really increase the value and impact of your paper. The section *"Summary"* could then be extended to *"Summary and outlook"* and hence summarise the perspectives of prospective research based on this dataset, and maybe how it could be combined with other datasets.

**Horizontal and vertical representativeness**

The supplemental campaigns are a great feature of this dataset, yet the importance of the campaign data is, in my opinion, not adequately represented. Only in ll. 364-367, you briefly address the issue of spatial representativeness of the permanent sensors, based on campaign data used by Benninga et al. (2020). However, that publication only looked into this issue for a small part of the fields, and also not in detail. I think it is worth providing a more comprehensive assessment of the uncertainty arising from a lack of horizontal representativeness, by using the campaign data. It does not need to be exhaustive, but at least I don't see why you cannot provide such an uncertainty estimate for all stations.

It is a bit unfortunate, of course, that the campaigns were limited to the upper 5 cm. It is of course more tedious, but still feasible to carry out snapshot campaigns for vertical profiles by successively drilling to the next measurement depth. With the data limited to the surface, it is impossible to assess horizontal and vertical representativeness in a uniform framework. I assume that the limitation of the campaigns was partly justified by resource constraints, and partly by the motivation to use the data for ground-truthing remotely sensed soil moisture products. However, it should not be me speculating… so I would like to ask the authors to be more transparent about that study design and its justification, as well as about the resulting limitations and uncertainties with regard to soil hydrological applications.

**Calibration**

I appreciate the effort put into the calibration of the probes by using gravimetric measurements. Still, I am wondering whether the presentation of the corresponding procedures and the results could be more concise.

Furthermore, I think that you usually calibrate the relationship between permittivity and soil moisture (which is non-linear), while you took the volumetric soil moisture reading and linearly "re-calibrated" them. Please elaborate briefly on possible implications, if this makes sense to you.

**Integration into ISMN**

Why are these data not integrated into the ISMN? I would be interested in hearing your opinion on this. Many of the stations included in the ISMN are very similar types of profile measurements, and it should be helpful to put your data into an immediate context with these.

**Homogeneity**

You pointed out that some of the monitoring locations had to be changed during operations. Is there any way to assess the effect on data homogeneity? Are there periods of overlap for the new and the old locations for such an assessment?

**Missing data**

Please specify, maybe in a table, the percentage of missing data for each station.

**Data repository**

I found the data repository (https://easy.dans.knaw.nl) very slow. Even for the small files, I sometimes had to wait for a long time for the download to start. I do not know the EASY repository, and it also does not offer much information. I assume it is ok, but maybe ESSD team members could check whether it meets the standards of ESSD? In the end, I could download the data, so it's fine for me.

**Other specific comments**

- *Introduction*: at the end of the introduction, could you briefly outline the structure of the manuscript, specifically since the sections are not related to a conventional manuscript structure?
- *Ll. 32*: Gravimetric measurements are also unsuitable for monitoring purposes due to invasive and destructive nature of the measurement
- *Ll. 36*: why "*relative* electric permittivity" ?
- *Ll. 73-80*: Why is this required? I suggest removing this paragraph.
- *Comments on maps in Fig. 1-3:* these should be made more clear and less redundant
    - In my view, Fig. 1 is not required as a standalone figure, it could be included as an inset map in the other maps

- Fig. 2: Why are specific land cover types removed from the DEM? Apart from that, the topography map could be combined with Fig. 3 (soil map), e.g. by using contour lines and/or hillshading
- Fig. 3: Please use hollow markers so the reader can better see the land use behind a location marker; the colour for sand is unnecessarily dominant which makes the figure kind of fidgety
- Fig. 4: again, the choice of colours is quite glaring. More importantly, though, the choice is not suited for readers with color vision deficiency.
- I would find it very helpful to have the monitoring sites and the precipitation gauges in one map. Precipitation is crucial to interpret the soil moisture observations, and it is unfortunate that there is not a closeby rain gauge for each profile probe. At least on the map, the reader should get an impression about the distance to the next rain gauge. Maybe that information could also be included in the location metadata?
- You might also mention the open radar composite data. This data also has its issues, but it provides full coverage and hence provide useful information specifically in the case of convective rainfall in the summer season (https://dataplatform.knmi.nl/dataset/radar-corr-accum-03h-1-0, https://dataplatform.knmi.nl/dataset/nl-rdr-data-rtcor-5m-1-0).
- *Ll. 153-155*: *"The instrumentation is [...] typically placed at the border of fields and preferably several tens of metres away from disturbing features (i.e. trees, roads or watercourses) [...] to minimize disturbance from recurring farming practices and optimize its representativeness for the adjacent fields."* How does the site selection, as described here, affect the representativeness for the agricultural fields? As pointed out above, it would be great if this issue was addressed in more detail in a dedicated section on uncertainty.
- *Ll. 157-159*: Are the site properties (specifically soil!) only derived from the soil map or were they also specifically determined for the exact point (texture, SOM, horizons, …)?
- *Table 1:* I think that the table is useful, but in its current form, it might also go to the supplement, specifically since parts of the legend (asterisks) are also explained there. I think that the column "soil type class" should rather be "texture class".
- *Ll. 169-171*: I think not all these details on sensor construction are required, please replace by adequate reference.
- *L. 238:* *"The sampling strategy during campaigns aimed at characterizing the top 5 cm soil moisture content of fields."* Stating the target variable (SM at 5 cm) is, in my view, not really a sampling strategy. Please also see my above comment on limiting the campaign to surface SM.
- *Section 4.1:* How was the position of the sampling locations measured (GPS or DGPS / accuracy)?
- *Ll. 238-240:* The criteria of selecting a sampling location remain unclear. Which role plays the distance between locations and the parcel size, as compared to a random placement? Fig. 8. does not really help me to understand the samling design.
- *Ll. 242-245:* Which design was used for locations that were neither grassland nor maize?
- *L. 246:* What do you mean by *"The collection of soil samples for GVSM determination [...] stopped [...]"*? Does that mean that after a specific campaign day, you decided that more gravimetric samples were not required?
- *Fig. 9:* I find the figure different to read because all the information is represented by the color code. Please consider a different approach: you could e.g represent the year by a

colored circle and the gravimetric measurement by a marker inside the circle. I would also use a different marker for the maize stems.

- *Fig. 10:* the labels are too small, the resolution insufficient.
- *L. 278: "[...] to develop the calibration functions for the ThetaProbe on a field campaign basis [...]"* - please be more specific about what you mean here.
- *Ll. 285-286:* The agreement for Fig. 11 seems higher, but it should be noted that the points are mainly from a different set of stations: so is the better agreement due to the probe, or due to the different locations? Of course, that is difficult to say, but maybe the issue could be briefly mentioned?
- *L. 286:* replace "little" by "small"
- *L. 287:* "distributions" is not the adequate term here, I think. Isn't it the relationships between probe readings and GVSM that you mean?
- *Ll. 296-297: "The performance metrics presented in Tables 4 and 5 show that the matching probe ('site') and GVSM measurements generally led to better performance except for the 2009 field campaign [...]"* - I don't agree entirely: in Tab. 5, the "mean" variant performs better for ITC_SM02.
- *Ll. 297-299: "Of the field campaign calibrations, the calibration developed for the HydraProbe (2016-2017) led to the best results with a RMSE of 0.032 $m^3$ $m^{-3}$ versus 0.041 $m^3$ $m^{-3}$ for 2015 and 0.048 $m^3$ $m^{-3}$ for 2009."* This sentence is ambiguous, please change to: *"Of the field campaign calibrations, the calibration developed for the HydraProbe (2016-2017) led to the best results with an RMSE of 0.032 $m^3$ $m^{-3}$ in comparison to RMSEs of 0.041 $m^3$ $m^{-3}$ for 2015 and 0.048 $m^3$ $m^{-3}$ for 2009, as obtained for the ThetaProbes."*
- L. 325: readme *file*
- *L. 347: "Long-term operation of in-situ monitoring networks goes hand in hand with measurement uncertainties."* - please remove this sentence.
- *Ll. 350-353:* I understand this is too late, but you could also recalibrate the relation between the prime measurement variable (raw voltage) and permittivity yourself, by using pure air and water, for each sensor - correct?
- *Table 8* should also contain, in a dedicated column, the main variable(s) represented by the corresponding third party datasets

**Technical comments**

- *Dataset:* In the files in `0_measurement_campaigns/1_processed_calibrated`, the combination of field IDs and location IDs are not unique. E.g. for file `Twente_fieldwork_ITCSM03_pd_cd.csv`, the first six rows show, for the field column, the values `1, 1, 1, 1, 1, 1` while the location column shows `1, 2, 3, 1, 2, 3` while I would have expected `1, 2, 3, 4, 5, 6`. Could you elaborate?
- The header of your data repository is "Ten years profile soil moisture and temperature measurements in Twente (version 2022)" - that is inconsistent with the title of your paper.
- *Ll. 22 ff.: "conversion of water into vapour via evapotranspiration at the expense of solar radiation"* - I find that weirdly phrased
- *Punctuation* requires revision throughout the paper
- *L. 114:* Is heath actually cultivated in that context? I found it surprising to list it under agricultural land use.
- *L. 175:* replace "circumstances" by "conditions"

- *L. 175:* replace "influence zone" by "footprint" or "spatial representativeness", if that is what is meant
- *Fig. 8:* the north arrow appears unnecessarily large
- *L. 283:* I suggest to remove a) and b) from the text, this is included in the figure caption
- *L. 405:* measurement**s**

---

## Author Comment (AC1)

**Authors' response to Referee 1**

In their manuscript "Twelve years profile soil moisture and temperature measurements in Twente, the Netherlands", R. van der Velde and colleagues present a corresponding dataset which is supplemented by several soil moisture measurement campaigns which add more points around the long-term measurements in terms of snapshots.

I congratulate the author team for a precious and useful dataset. The overall approach and design (a sparse network of profile probes) is not specifically innovative, yet it allows for interesting analyses and should be a useful asset for the scientific community. Most importantly, the author spent a lot of effort in the (re-)calibration of the soil moisture sensors by comparing the instrumental readings to gravimetric soil moisture measurements – an effort that is well-spent, and certainly nothing to be taken for granted.

The paper itself is mostly well-written, the accompanying dataset well-organised, complete, and contains both raw readings and processed data, conforming with various user requirements.

I still have some specific comments which you might classify as major, but which I am confident the authors will be able to consider in one way or the other. Most of them should be understood as suggestions to improve the manuscript.

Authors' response:
We thank the referee for the positive and detailed comments. The level of detail and provided suggestions are much appreciated and motivate us to critically reflect at the content of the submitted manuscript. In the text below, we provide our point-by-point response to the comments.

Most of the referee's suggestions have been implemented. We are currently working on a few remaining referee comments that require additional research or restructuring of the manuscript. Our response to the referee comments is structured as follows:
- The native referee comment is labelled and written in black.
- The authors response is written in blue
- Text from the manuscript is written with the Times New Roman font whereby the native text is in black and the changes are in red.

**General comments**
**R1GC1 (Referee 1 General Comment 1): Structure and balance of the manuscript**
I think in the final parts of the manuscript, the structure should be adjusted to make it more straightforward. My suggestion is to drop section "5.1 Maintenance monitoring stations" (which is, in my opinion, not relevant in the context of this paper), and to combine section "5.2 Data processing and flagging" with the section "Data availability" (please note that the section numbering is wrong after section 5). Section "5.3 Known issues" deserves, in my view, a dedicated single subsection, but should be renamed to "Data uncertainty". Please use this section to provide a more comprehensive synthesis on the two most important sources of uncertainty: the sensor calibration (most importantly the relationship between permittivity and volumetric soil moisture, but maybe also some words on the relationship between the primary measurement variable, typically raw voltage, and permittivity, which is assumed to be set by the

manufacturer), and the spatial representativeness (see comment below). The section title "Time series" is not really adequate. Still, I think it is an important section as it presents an opportunity to highlight the scientific potential of your dataset (see comment "Underlying research questions and prospective application of the data").

Apart from the structure, parts of the paper could be more concise. Please reconsider whether some details on the instruments' construction design and the calibration procedure could be removed for the sake of readability.

Authors' response:
Thank you for the suggestion for restructuring section 5, which we are in the process of completing. In the revision so far, we have dropped section 5.1, renamed section 5 Data uncertainties with subsections 5.1 sensor calibration and 5.2 spatial representativeness. In the revised section 5.1 we discuss the sensor calibrations, the relationships between the raw measurements, relative electric permittivity and soil moisture. In section 5.2 we discuss the spatial representativeness through comparisons between the station measurements and measurements collected during the field campaigns at different spatial scales. This will provide readers and prospective data users with idea on how to best use the data.

Further, we have critically reviewed the text and presentations of manuscript, by shortening descriptions, removing redundancies and paragraphs with less relevance to the context of the manuscript.

**R1GC2 Underlying research questions and prospective application of the data**
In ll. 44-49, you describe previous research that was based on the presented dataset, and which already started more than 10 years ago. While this exemplifies potential use cases of the dataset, it also leaves the reader with a feeling that this dataset was only published after it had already been comprehensively used.

Of course, that is not true! All the more, however, it is important for this paper to highlight and to exemplify open research questions and application cases related to this data. I think the section that is now entitled "Time series" is a good opportunity to highlight interesting properties of your dataset - as you already started to do. Still, I am wondering why you only show 3 out of 20 stations and 4 out of 12 years. Of course, this paper should not be about a scientific analysis, and of course you should not just show time series of all stations at all depths for all years. But maybe you could demonstrate the potential of your dataset along some exemplary research questions, maybe also at different time scales? This is just a thought, but I think this section could really increase the value and impact of your paper. The section "Summary" could then be extended to "Summary and outlook" and hence summarise the perspectives of prospective research based on this dataset, and maybe how it could be combined with other datasets.

Authors' response:
In the revised manuscript we have renamed the section 'time series' to 'research opportunities' and intend to discuss in this section research topics, such as
-   the validation of both satellite and model-based soil moisture products,
-   the development of retrieval algorithms and process models with an agro-hydrological and hydrometeorological focus,

- the translation of surface to rootzone soil moisture and links with groundwater,
- the analysis of infiltration processes and root zone soil moisture behaviour under water limiting conditions,
- the analysis of spatio-temporal scalability using a combination of the soil moisture measurements collected during the field campaigns and at the stations. Such analysis would complement other analyses that have done in the past with soil moisture dataset that have been collected in other parts of the world.

As the referee suggests, the summary section of the revised manuscript will be extended to a 'summary and outlook' to include also the perspectives on prospective research.

**R1GC3 Horizontal and vertical representativeness**
The supplemental campaigns are a great feature of this dataset, yet the importance of the campaign data is, in my opinion, not adequately represented. Only in ll. 364-367, you briefly address the issue of spatial representativeness of the permanent sensors, based on campaign data used by Benninga et al. (2020). However, that publication only looked into this issue for a small part of the fields, and also not in detail. I think it is worth providing a more comprehensive assessment of the uncertainty arising from a lack of horizontal representativeness, by using the campaign data. It does not need to be exhaustive, but at least I don't see why you cannot provide such an uncertainty estimate for all stations. It is a bit unfortunate, of course, that the campaigns were limited to the upper 5 cm. It is of course more tedious, but still feasible to carry out snapshot campaigns for vertical profiles by successively drilling to the next measurement depth. With the data limited to the surface, it is impossible to assess horizontal and vertical representativeness in a uniform framework. I assume that the limitation of the campaigns was partly justified by resource constraints, and partly by the motivation to use the data for ground-truthing remotely sensed soil moisture products. However, it should not be me speculating… so I would like to ask the authors to be more transparent about that study design and its justification, as well as about the resulting limitations and uncertainties with regard to soil hydrological applications.

Authors' response:
The reviewer is correct by assuming that the design of the field campaign data was motivated by the use of the data for the validation of remote sensing products. With this objective in mind, resources were focused on the sampling of a large spatial extent rather than collecting soil moisture profiles across fields. We have clarified this in the revised manuscript.

In the revised manuscript we will also report on comparisons between the field campaign measurements and the station data. We will reported and discuss uncertainty levels on stations basis and investigate whether general conclusions can be draw based on land cover.

**R1GC4 Calibration**
I appreciate the effort put into the calibration of the probes by using gravimetric measurements. Still, I am wondering whether the presentation of the corresponding procedures and the results could be more concise. Furthermore, I think that you usually calibrate the relationship between permittivity and soil moisture (which is non-linear), while you took the volumetric soil moisture reading and linearly "re-calibrated" them. Please elaborate briefly on possible implications, if this makes sense to you.

Authors' response:
During the revision process, we shortened the descriptions of both the applied procedure and results, and removed figures with redundant information.

In the new subsection 5.1 *Sensor calibration* of the section 5 *Data uncertainties* we have devoted the first two paragraphs to the possible implications of linearly calibrating the probe VSM against GVSM. The reason for linearly re-calibrating the native volumetric soil moisture content reading was to remain consistent with the earlier EC-TM calibration reported by Dente et al. (2011).

METER group has adopted third order polynomials to describe the relationship between VSM and the $\varepsilon_r$ for the 5TM (Topp et al. 1980) and EC-TM (equivalent to the Topp equation) probes. The ThetaProbe and HydraProbe use a linear relationship between the VSM and the refractive index ($\sqrt{\varepsilon_r}$), which can be rewritten as a second order polynomial. When we use these equations and apply the calibration coefficients the manufacturers provide for mineral soil, the relationships of $\varepsilon_r$ versus VSM are obtained as shown in Figure R1.

[Figure]

Figure R1: $\varepsilon_r$ against VSM using the mineral soil calibrations provided for the EC-TM, 5TM, ThetaProbe and HydraProbe.

The relationships shown in Figure R1 have similar shapes. To explore this further, Figure R2 shows the EC-TM, ThetaProbe and HydraProbe VSM against the 5TM VSM. Indeed the relationships in Figure R2 are close to linear.

[Figure]

Figure R2: EC-TM, ThetaProbe and HydraProbe VSM against the 5TM VSM.

We fit linear equations through data points of probe (EC-TM, ThetaProbe and HydraProbe) VSM and 5TM VSM to mimic the VSM and GVSM calibration procedure reported in the manuscript. The VSM difference between the EC-TM, ThetaProbe and HydraProbe VSM and the 5TM VSM matched through application of the established linear fits is shown in Figure R3.

Figure R3 shows that the differences are not negligible with maximum values up to 0.025 $m^3\,m^{-3}$ in the wet limit for the ThetaProbe and HydraProbe. This is the consequence of the difference between the second and third order polynomial. A summary of these results are included in section 5.1.

We expect, however, that the overall implications will be limited because in the limit with SM<=0.5 $m^3\,m^{-3}$ a linear relationship is well established between the VSM readings of the probe and the independently measured GVSM. Other sources of uncertainties, such as spatial scale mismatch and sampling errors, are apparently dominant over the ambiguity caused by the shape of the VSM and $\varepsilon_r$ relationship. Further investigation of the form of the relationship between the VSM and $\varepsilon_r$ goes beyond the scope of the manuscript and requires different experimental setup.

[Figure]

Figure R3: VSM difference between the EC-TM, ThetaProbe and HydraProbe VSM and the 5TM VSM matched through application of established linear fits.

**R1GC5 Integration into ISMN**
Why are these data not integrated into the ISMN? I would be interested in hearing your opinion on this. Many of the stations included in the ISMN are very similar types of profile measurements, and it should be helpful to put your data into an immediate context with these.

Authors' response:
In future we will integrate the data into the ISMN. The reason why this has not been done yet is the availability of time and the choice that our university has made to use DANS-easy  as trusted repository, which offers a guaranteed permanent DOI.

**R1GC6 Homogeneity**
You pointed out that some of the monitoring locations had to be changed during operations. Is there any way to assess the effect on data homogeneity? Are there periods of overlap for the new and the old locations for such an assessment?

Authors' response:

Unfortunately the change of the locations had to happen most of the time on an ad hoc basis. This has been the consequence of using privately owned land for monitoring purposes. In one case, (2017 change of ITC_SM03), we had the opportunity to have instrumentation operational at two location along the field. However, the period of overlap was found too short (two months) for the data to be used to assess the data homogeneity.

**R1GC7 Missing data**
Please specify, maybe in a table, the percentage of missing data for each station.

Authors' response:
We have added the percentage missing data to Table S1 for each station. The missing data is on average 13.5%.

**R1GC8 Data repository**
I found the data repository (https://easy.dans.knaw.nl) very slow. Even for the small files, I sometimes had to wait for a long time for the download to start. I do not know the EASY repository, and it also does not offer much information. I assume it is ok, but maybe ESSD team members could check whether it meets the standards of ESSD? In the end, I could download the data, so it's fine for me.

Authors' response:
We apologize for this technical hiccup. We were informed by our data steward that at the time the referee downloaded the data the speed may have been slower due to the migration of the system.

Indeed the data repository is slow, especially when one wants to access many files at once. This was especially the case in the 2021 version of the dataset. The 2022 version is reduced to 8 files with sizes varying from 1.6 MB up to 1.6 GB. When the files are downloaded one at the time the retrieval of the data via DANS-easy is doable, but of course it take more time to download a large file than a smaller one.

**Specific comments:**
**R1SC1:**
- Introduction: at the end of the introduction, could you briefly outline the structure of the manuscript, specifically since the sections are not related to a conventional manuscript structure?

Authors' response:
We will add a paragraph to introduction that describes the structure of the manuscript.

**R1SC2:**
- Ll. 32: Gravimetric measurements are also unsuitable for monitoring purposes due to invasive and destructive nature of the measurement

Authors' response:
We have modified the text as follows:

Gravimetrically determined soil moisture measurements are, however, destructive in nature and labor intensive. The gravimetric approach is as such unsuitable for monitoring purpose due to inherent limitation

in collecting repetitive measurements and has also become unfeasible for long-term monitoring as the cost of labor increased.

**R1SC3:**
- Ll. 36: why "relative electric permittivity" ?

Authors' response:
We assume that the referee would like see a motivation why the relative electric permittivity measurements are an appropriate technique for *in-situ* soil moisture monitoring. We have modified the text as follows:

Therefore, indirect estimation of the soil water content has been widely investigated (e.g. Vereecken et al. 2008). The devices that measure the soil's relative electric permittivity ($\varepsilon_r$) are nowadays commonly used for regional scale soil moisture monitoring networks (e.g. Martinez-Fernandez and Cebalos 2005, Calvet et al. 2007, Su et al. 2011, Bircher et al. 2012, Smith et al. 2012, Benninga et al. 2018, Bogena et al. 2018, Caldwell et al. 2019, Tetlock et al. 2019) because of the strong and direct relationship between the $\varepsilon_r$ and soil moisture as a result of the large contrast between the $\varepsilon_r$ of dry soil (3-5) and water (80) at frequencies lower than about 1 GHz.

**R1SC4:**
- Ll. 73-80: Why is this required? I suggest removing this paragraph.

Authors' response:
We have removed the paragraph.

**R1SC5:**
- Comments on maps in Fig. 1-3: these should be made more clear and less redundant

**R1SC6:**
- In my view, Fig. 1 is not required as a standalone figure, it could be included as an inset map in the other maps

**R1SC7:**
- Fig. 2: Why are specific land cover types removed from the DEM? Apart from that, the topography map could be combined with Fig. 3 (soil map), e.g. by using contour lines and/or hillshading

**R1SC8**
- Fig. 3: Please use hollow markers so the reader can better see the land use behind a location marker; the colour for sand is unnecessarily dominant which makes the figure kind of fidgety.

**R1SC9:**
- Fig. 4: again, the choice of colours is quite glaring. More importantly, though, the choice is not suited for readers with color vision deficiency.

**R1SC10:**
- I would find it very helpful to have the monitoring sites and the precipitation gauges in one map. Precipitation is crucial to interpret the soil moisture observations, and it is unfortunate that there is not a closeby rain gauge for each profile probe. At least on the map, the reader

should get an impression about the distance to the next rain gauge. Maybe that information could also be included in the location metadata?

Authors' response:
We agree with the above comments of the reviewer and replaced the Figs. 1 -4 with one single figure (Figure R4) based on the DEM showing the locations of the soil moisture stations, weather stations, precipitation stations and groundwater monitoring wells with different symbols. The land cover and soil map are excluded from the manuscript for the reason that the limited spatial variability shown in the maps is also described in the text.

[Figure]

Figure R4.

**R1SC11:**
- You might also mention the open radar composite data. This data also has its issues, but it provides full coverage and hence provide useful information specifically in the case of convective rainfall in the summer season (https://dataplatform.knmi.nl/dataset/radar-corr-accum-03h-1-0, https://dataplatform.knmi.nl/dataset/nl-rdr-data-rtcor-5m-1-0).

Authors' response:
The following sentence has been added to the end of the first paragraph of section 2.4

In addition, radar-derived precipitation is available as approximately 1 km gridded files for the Netherlands as gauge corrected accumulations for 5 min, 3 and 24 hours.

And a row has been added to Table 8 providing data access instructions.

**R1SC12:**

- Ll. 153-155: "The instrumentation is [...] typically placed at the border of fields and preferably several tens of metres away from disturbing features (i.e. trees, roads or watercourses) [...] to minimize disturbance from recurring farming practices and optimize its representativeness for the adjacent fields." How does the site selection, as described here, affect the representativeness for the agricultural fields? As pointed out above, it would be great if this issue was addressed in more detail in a dedicated section on uncertainty.

Authors' response:
This is indeed an important and interesting question. In the revised manuscript we will address the issue of the representativeness using the field campaign measurements as the referee suggests here and also in comment R1GC3. The sentence referred to in this comments has been modified as follows,

The instrumentation is, therefore, typically placed at the border of fields and preferably several tens of metres away from disturbing features (i.e. trees, roads or watercourses), as shown in Fig. 6, to minimize disturbance from recurring farming practices and optimize its representativeness for the adjacent fields, which is further addressed in section 5.2.

**R1SC13:**
- Ll. 157-159: Are the site properties (specifically soil!) only derived from the soil map or were they also specifically determined for the exact point (texture, SOM, horizons, …)?

Authors' response:
These properties are derived from the soil map and have not been determined in the soil laboratory. In the revised manuscript we have clarified this by modifying the text as follows,

Table 1 lists for each station the texture class derived from the soil map, the land cover per year of the adjacent fields and the maintenance operations carried out.

**R1SC14:**
- Table 1: I think that the table is useful, but in its current form, it might also go to the supplement, specifically since parts of the legend (asterisks) are also explained there. I think that the column "soil type class" should rather be "texture class".

Authors' response:
We have migrated the table to the supplement.

**R1SC15**
- Ll. 169-171: I think not all these details on sensor construction are required, please replace by adequate reference.

Authors' response:
In the revised manuscript we refer readers to the 5TM and EC-TM manuals for these details and added the following sentence to the end of the paragraph,

'Readers are referred the manuals of the manufacturer for the details on the instrument design and the specifications(Decagon Devices 2008 and 2017).'

**R1SC16**

- L. 238: "The sampling strategy during campaigns aimed at characterizing the top 5 cm soil moisture content of fields." Stating the target variable (SM at 5 cm) is, in my view, not really a sampling strategy. Please also see my above comment on limiting the campaign to surface SM.

Authors' response:
We have modified the sentence to read,

The sampling strategy during campaigns was designed to validate soil moisture retrievals from satellite observations for which the top 5 cm soil moisture content was measured within fields.

**R1SC17**
- Section 4.1: How was the position of the sampling locations measured (GPS or DGPS / accuracy)?

Authors' response:
The accuracy of the position of the sampling points was not something that we recorded, but from our experience the GPS accuracy in our study area is and was 3 -4 m. The following sentence has been added at the end of the 1st paragraph.

Measurement locations have been determined using GPS with an accuracy better than 4 meters.

**R1SC18**
- Ll. 238-240: The criteria of selecting a sampling location remain unclear. Which role plays the distance between locations and the parcel size, as compared to a random placement?

Authors' response:
The sentence has been reformulated to read,

A maximum of six measurement locations were selected per field about 50 m to 100 m apart, which was reduced to a minimum of three locations when the size of parcel was not big enough.

**R1SC19**
Fig. 8. does not really help me to understand the sampling design.

Authors' response:
We agree with the reviewer that the figure is of little added value in explaining the sampling design. We have, therefore, removed the figure from the manuscript.

**R1SC20**
- Ll. 242-245: Which design was used for locations that were neither grassland nor maize?

Authors' response:
We agree with the referee that the description is incomplete and modified the text as follows:

At fields without crop rows, such as grass and wheat, soil moisture was measured with the impedance probe at four to nine points within a 1 m$^2$ plot and next to one of the probe readings a soil sample was taken for GVSM determination. In fields with crop rows, such as maize and potato, probe readings were taken along the transect perpendicular to the crop rows, approximately 0.75 m apart, with the soil sample taken in between two rows.

**R1SC21**
- L. 246: What do you mean by "The collection of soil samples for GVSM determination [...] stopped [...]"? Does that mean that after a specific campaign day, you decided that more gravimetric samples were not required?

Authors' response:
It is exactly that!

During field campaigns the collection soil samples is the most time-consuming activity in the field, and also the 'post-processing', weighing and oven-drying of soil samples, requires a lot of attention. Every season we have collected pairs of impedance probe recordings and soil samples. Throughout the campaigns we monitored the impedance probe – GVSM relationship that was collected and ended the soil sampling when we were satisfied or when additional sampling would be of little added value due to similar soil moisture conditions. An exact of number of independent samples is difficult to attach to this (but ideally > 25) because the cut-off would also depend on the dynamic soil moisture range covered up to that moment, and how many days the same conditions had been sampled.

We modify the sentence as follows,

The collection of soil samples for GVSM determination was done each field campaign to calibrate the probe readings and stopped when the covered dynamic range and number of matchups, ideally greater than 25, were suitable to establish a calibration function.

**R1SC22**
- Fig. 9: I find the figure different to read because all the information is represented by the color code. Please consider a different approach: you could e.g represent the year by a colored circle and the gravimetric measurement by a marker inside the circle. I would also use a different marker for the maize stems.

Authors' response:
We have used these suggestions to replace the previous figure by the one below.

[Figure]

Figure R5

**R1SC22 here**
- Fig. 10: the labels are too small, the resolution insufficient.

Authors' response:
We have reproduced the figure (figure 9 in the revised manuscript) with a higher resolution. At the same time, we taken the opportunity to improve its suitability for readers with colour vision deficiency by using different symbols and colour tones. Figures 7, 10 and 11 have also been reproduced in the same style for the revised manuscript.

[Figure]

Figure R6

**R1SC23**

- L. 278: "[...] to develop the calibration functions for the ThetaProbe on a field campaign basis [...]" - please be more specific about what you mean here.

Authors' response:
We have modified this sentence as follows,

Therefore, we have chosen to develop the ThetaProbe calibration functions for the 2009 and 2015 field campaigns separately and not for individual stations or specific soil types.

**R1SC24**

- Ll. 285-286: The agreement for Fig. 11 seems higher, but it should be noted that the points are mainly from a different set of stations: so is the better agreement due to the probe, or due to the different locations? Of course, that is difficult to say, but maybe the issue could be briefly mentioned?

Authors' response:
We have added to following two sentences to briefly address this issue,

Factors that could have contributed to this agreement difference are the deployed instruments, the different sets of fields sampled, the number of matchups collected per field and the extent of the dynamic soil moisture range covered by the matchups. However, it is beyond the scope of this manuscript to quantify their relative contributions.

**R1SC25**
- L. 286: replace "little" by "small"

Authors' response:
done

**R1SC26**
- L. 287: "distributions" is not the adequate term here, I think. Isn't it the relationships between probe readings and GVSM that you mean?

Authors' response:
We have modified the sentence as follows:

Also noticeable in Figs. 10 and 11 are the small differences among the apparent relationships represented by the clusters of data points belonging to individual stations, which again may question the added value of station-specific calibration functions.

**R1SC27**
- Ll. 296-297: "The performance metrics presented in Tables 4 and 5 show that the matching probe ('site') and GVSM measurements generally led to better performance except for the 2009 field campaign [...]" - I don't agree entirely: in Tab. 5, the "mean" variant performs better for ITC_SM02.

Authors' response:
The referee is correct in the sense that RMSE of mean variant performs better for ITC_SM02, when comparing the $R^2$ in the tables the statement does hold. We have modified the sentence and add one sentence as given below.

The performance metrics presented in Tables 4 and 5 show that the matching probe ('site') and GVSM measurements led to a better performance in terms of the $R^2$ except for the 2009 field campaign. The same holds when comparing the RMSEs with exception of the 2016-2017 results for ITC_SM02 in which case the mean of the probe readings leads to a better performance.

**R1SC28**
- Ll. 297-299: "Of the field campaign calibrations, the calibration developed for the HydraProbe (2016-2017) led to the best results with a RMSE of 0.032 m3 m-3 versus 0.041 m3 m-3 for 2015 and 0.048 m3 m-3 for 2009." This sentence is ambiguous, please change to: "Of the field campaign calibrations, the calibration developed for the HydraProbe (2016-2017) led to the best results with an RMSE of 0.032 m3 m-3 in comparison to RMSEs of 0.041 m3 m-3 for 2015 and 0.048 m3 m-3 for 2009, as obtained for the ThetaProbes."

Authors' response:

Thanks, we have modified the sentence as suggested.

**R1SC29**
- L. 325: readme file

Authors' response:
'file' has been added

**R1SC30**
- L. 347: "Long-term operation of in-situ monitoring networks goes hand in hand with measurement uncertainties." - please remove this sentence.

Authors' response:
Done

**R1SC31**
- Ll. 350-353: I understand this is too late, but you could also recalibrate the relation between the prime measurement variable (raw voltage) and permittivity yourself, by using pure air and water, for each sensor - correct?

Authors' response:
This is less straightforward than one would expect. The raw sensor output of the METER group probes are not voltages but digital numbers produced with the internal firmware from the prime observations. In the case of the 5TM probe, dividing by 50 returns the electric permittivity, but in the case of the EC-TM probe, this is not clear.

Further, a linear response between the voltage and the permittivity need to be assumed. The range from 1 (air) up to 80 (water) is too large for this assumption and ideally several additional standards would be used to establish such relationship as it done at METER Group (formerly Decagon). Via personal communication (see copy of e-mail below) METER Group informed us that they use IPA with a permittivity of ~40 as the highest calibration standard for their sensor. This probably also links to the specified accuracy of the permittivity measurements +/- 1 below a permittivity of 40 and 15% above a permittivity of 40 (see link). This non-linearity is also described in a passage that we can across in the EC-TM manual (see link).

[Figure]

Prior to the deployment of the instruments, we would indeed verify the functioning of the probes using air and water, and found for the 5TM that in air the reading would always be

equivalent to a permittivity of 1, but in water reading would be less constant in time and also a sensor-to-sensor variability would be noticeable as was previously studied by Rosenbaum et al. (2010).

The above described detailed are included in the newly written section 5.1.

Rosenbaum, U., Huisman, J. A., Weuthen, A., Vereecken, H., and Bogena, H. R.: Sensor-to-Sensor Variability of the ECH2O EC-5, TE, and 5TE Sensors in Dielectric Liquids, Vadose Zo. J., 9, 181–186, https://doi.org/10.2136/vzj2009.0036, 2010.

**R1SC32**
- Table 8 should also contain, in a dedicated column, the main variable(s) represented by the corresponding third party datasets

Authors' response:
We will add such column to the table.

**Technical comments**
- Dataset: In the files in 0_measurement_campaigns/1_processed_calibrated, the combination of field IDs and location IDs are not unique. E.g. for file Twente_fieldwork_ITCSM03_pd_cd.csv, the first six rows show, for the field column, the values 1, 1, 1, 1, 1, 1 while the location column shows 1, 2, 3, 1, 2, 3 while I would have expected 1, 2, 3, 4, 5, 6. Could you elaborate?

Authors' response:
Yes the reviewer is correct. This is an artefact from the way in which we recorded the measurements during fieldwork. In the revised dataset we have corrected this.

- The header of your data repository is "Ten years profile soil moisture and temperature measurements in Twente (version 2022)" - that is inconsistent with the title of your paper.

Authors' response:
This is indeed a bit annoying. We will contact DANS-EASY try to fix it.

- Ll. 22 ff.: "conversion of water into vapour via evapotranspiration at the expense of solar radiation" - I find that weirdly phrased

Authors' response:
We have changed this sentence to,

Moreover, the availability of soil moisture for evapotranspiration governs heat and water exchanges between the land surface and atmosphere impacting weather and climate

- Punctuation requires revision throughout the paper.

Authors' response:
We have carefully checked the punctuation throughout the manuscript

- L. 114: Is heath actually cultivated in that context? I found it surprising to list it under agricultural land use.

Authors' response:
That is correct. We have removed 'heath' as well as 'forest'.

- L. 175: replace "circumstances" by "conditions"

Authors' response:
Done

- L. 175: replace "influence zone" by "footprint" or "spatial representativeness", if that is what is meant

Authors' response:
We have reformulated the sentence as given below because footprint may be confused with the footprint of remote sensing instruments and 'spatial representativeness' is straightforward to define in the soil moisture context.

'Benninga et al. (2018) have shown under laboratory conditions that 5TM probe samples about 3 cm to 4 cm soil around the prongs.'

- Fig. 8: the north arrow appears unnecessarily large

Authors' response:
We will reproduce the figure with a smaller north arrow.

- L. 283: I suggest to remove a) and b) from the text, this is included in the figure caption

Authors' response:
Done

- L. 405: measurements

Authors' response
Change to 'measurement locations'

---

## Author Comment (AC2)

**Authors' response to Referee 2**

This paper presents a dataset of ongoing in situ soil moisture measurements in a region in the eastern part of the Netherlands. The dataset covers the time period since 2009, 20 locations, and measurements at five different depths (in general). The paper presents also results from field campaigns that resulted in calibration functions for the soil moisture sensors.

Authors' response:
We would like to thank the referee for general positive feedback. In the text below, we provide our point-by-point response to the comments.

Most of the referee's suggestions have been implemented. Our response to the referee comments is structured as follows:

- The native referee comment is labelled and written in black.
- The authors response is written in blue
- Text from the manuscript is written with the Times New Roman font whereby the native text is in black and the changes are in red

**R2GC1:**
The paper is in general sound and I have only minor comments. However, a major concern is whether publication is warranted in a high impact journal like ESSD, for only 20 point soil moisture time series for a period of 13 years. Question is whether the dataset is unique enough for this journal. I have suggested "minor revision", but alternatively a recommendation could also be "rejection". I leave this decision to the editor.

Authors' response:
Our paper presents a unique dataset of in-situ soil moisture measurements in the Eastern part of the Netherlands. The uniqueness of this data set stems from its design as a network of fixed stations covering an area of 45 km by 40 km and measuring the soil moisture profile at nominal depths 5 cm, 10 cm, 20 cm, 40 cm and 80 cm for more than 10 years. Most of the current scientific articles reporting datasets based on in-situ measurements have generally a smaller temporal and spatial coverage in comparison to the presented work as will be further motivated in the following paragraph. Our paper also reports on complementary spatially distributed field measurements collected during campaigns organized in four different years, namely 2009, 2015, 2016 and 2017. Moreover, we have made an effort to describe in this paper an extensive collection of open third-party datasets (i.e. land cover/use, soil information, elevation, groundwater and meteorological observations) that can support the use of the Twente soil moisture and temperature datasets by other scientists and professional.

We have further investigated the suitability of our manuscript for the ESSD by searching in the title for the keyword 'soil moisture' and found 31 articles. From this collection of articles, we could identify at least eleven contributions of which the dataset relied on in-situ measurements. Those are listed below. The datasets reported in all articles cover a time span not significantly more 10 years and a similar spatial extent as the Twente region, with exception of Bogena et al. (2022) who report on an European network of cosmic ray probes. Three out of the eleven identified articles have a topic comparable to our manuscript, which are Benninga et al. (2018), Tetlock et al. (2019) and Zhang et al.

(2021). We have looked up the number of citations in Web of Science for those three articles and found that they are cited 18, 23 and 15 times, respectively. All three have been cited more than the journals impact factor. This leads us to the conclusion that the impact of the topic of our manuscript is at the appropriate level for the ESSD journal.

*Acticles on in situ soil moisture measurement dataset:*

1. Bam, E. K. P., Brannen, R., Budhathoki, S., Ireson, A. M., Spence, C., and van der Kamp, G.: Meteorological, soil moisture, surface water, and groundwater data from the St. Denis National Wildlife Area, Saskatchewan, Canada, Earth Syst. Sci. Data, 11, 553–563, https://doi.org/10.5194/essd-11-553-2019, 2019.

*2. Benninga, H.-J. F., Carranza, C. D. U., Pezij, M., van Santen, P., van der Ploeg, M. J., Augustijn, D. C. M., and van der Velde, R.: The Raam regional soil moisture monitoring network in the Netherlands, Earth Syst. Sci. Data, 10, 61–79, https://doi.org/10.5194/essd-10-61-2018, 2018. (Web of Science citations: 18)*

3. Bogena, H. R., Schrön, M., Jakobi, J., Ney, P., Zacharias, S., Andreasen, M., Baatz, R., Boorman, D., Duygu, M. B., Eguibar-Galán, M. A., Fersch, B., Franke, T., Geris, J., González Sanchis, M., Kerr, Y., Korf, T., Mengistu, Z., Mialon, A., Nasta, P., Nitychoruk, J., Pisinaras, V., Rasche, D., Rosolem, R., Said, H., Schattan, P., Zreda, M., Achleitner, S., Albentosa-Hernández, E., Akyürek, Z., Blume, T., del Campo, A., Canone, D., Dimitrova-Petrova, K., Evans, J. G., Ferraris, S., Frances, F., Gisolo, D., Güntner, A., Herrmann, F., Iwema, J., Jensen, K. H., Kunstmann, H., Lidón, A., Looms, M. C., Oswald, S., Panagopoulos, A., Patil, A., Power, D., Rebmann, C., Romano, N., Scheiffele, L., Seneviratne, S., Weltin, G., and Vereecken, H.: COSMOS-Europe: a European network of cosmic-ray neutron soil moisture sensors, Earth Syst. Sci. Data, 14, 1125–1151, https://doi.org/10.5194/essd-14-1125-2022, 2022.

4. Fersch, B., Francke, T., Heistermann, M., Schrön, M., Döpper, V., Jakobi, J., Baroni, G., Blume, T., Bogena, H., Budach, C., Gränzig, T., Förster, M., Güntner, A., Hendricks Franssen, H.-J., Kasner, M., Köhli, M., Kleinschmit, B., Kunstmann, H., Patil, A., Rasche, D., Scheiffele, L., Schmidt, U., Szulc-Seyfried, S., Weimar, J., Zacharias, S., Zreda, M., Heber, B., Kiese, R., Mares, V., Mollenhauer, H., Völksch, I., and Oswald, S.: A dense network of cosmic-ray neutron sensors for soil moisture observation in a highly instrumented pre-Alpine headwater catchment in Germany, Earth Syst. Sci. Data, 12, 2289–2309, https://doi.org/10.5194/essd-12-2289-2020, 2020.

5. Godsey, S. E., Marks, D., Kormos, P. R., Seyfried, M. S., Enslin, C. L., Winstral, A. H., McNamara, J. P., and Link, T. E.: Eleven years of mountain weather, snow, soil moisture and streamflow data from the rain–snow transition zone – the Johnston Draw catchment, Reynolds Creek Experimental Watershed and Critical Zone Observatory, USA, Earth Syst. Sci. Data, 10, 1207–1216, https://doi.org/10.5194/essd-10-1207-2018, 2018.

6. Heistermann, M., Bogena, H., Francke, T., Güntner, A., Jakobi, J., Rasche, D., Schrön, M., Döpper, V., Fersch, B., Groh, J., Patil, A., Pütz, T., Reich, M., Zacharias, S., Zengerle, C., and Oswald, S.: Soil moisture observation in a forested headwater catchment: combining a dense cosmic-ray neutron sensor network with roving and hydrogravimetry at the TERENO site Wüstebach, Earth Syst. Sci. Data, 14, 2501–2519, https://doi.org/10.5194/essd-14-2501-2022, 2022.

7. Jackisch, C., Germer, K., Graeff, T., Andrä, I., Schulz, K., Schiedung, M., Haller-Jans, J., Schneider, J., Jaquemotte, J., Helmer, P., Lotz, L., Bauer, A., Hahn, I., Šanda, M., Kumpan, M., Dorner, J., de Rooij, G., Wessel-Bothe, S., Kottmann, L., Schittenhelm, S., and Durner, W.: Soil moisture and matric potential – an open field comparison of sensor systems, Earth Syst. Sci. Data, 12, 683–697, https://doi.org/10.5194/essd-12-683-2020, 2020.

8. Roche, J. W., Rice, R., Meng, X., Cayan, D. R., Dettinger, M. D., Alden, D., Patel, S. C., Mason, M. A., Conklin, M. H., and Bales, R. C.: Climate, snow, and soil moisture data set for the Tuolumne and Merced river watersheds, California, USA, Earth Syst. Sci. Data, 11, 101–110, https://doi.org/10.5194/essd-11-101-2019, 2019.

9. Schaffitel, A., Schuetz, T., and Weiler, M.: A distributed soil moisture, temperature and infiltrometer dataset for permeable pavements and green spaces, Earth Syst. Sci. Data, 12, 501–517, https://doi.org/10.5194/essd-12-501-2020, 2020.

*10 Tetlock, E., Toth, B., Berg, A., Rowlandson, T., and Ambadan, J. T.: An 11-year (2007–2017) soil moisture and precipitation dataset from the Kenaston Network in the Brightwater Creek basin, Saskatchewan, Canada, Earth Syst. Sci. Data, 11, 787–796, https://doi.org/10.5194/essd-11-787-2019, 2019. (Web of Science citations: 23)*

*11. Zhang, P., Zheng, D., van der Velde, R., Wen, J., Zeng, Y., Wang, X., Wang, Z., Chen, J., and Su, Z.: Status of the Tibetan Plateau observatory (Tibet-Obs) and a 10-year (2009–2019) surface soil moisture dataset, Earth Syst. Sci. Data, 13, 3075–3102, https://doi.org/10.5194/essd-13-3075-2021, 2021. (Web of Science citations: 15)*

**R2SC1:**
L28. Instead of Mecklenburg et al., 2016 an earlier citation should be included.

Authors' response:

The citation Mecklenburg et al. (2016) has been replace to a citation to Kerr et al. (2010)

Kerr, Y.H., Waldteufel, P., Wigneron, J.-P., Delwart, S., Cabot, F., Boutin, J., Escorihuela, M.-J., Font, J., Reul, N., Gruhier, C., Juglea, S. E., Drinkwater, M.R., Hahne, A., Martin-Neira, M., and Mecklenburg, S.: The SMOS mission: New tool for monitoring key elements of the global water cycle, P. IEEE, 98, 666-687, doi: 10.1109/JPROC.2010.2043032, 2010.

**R2SC2:**
L114. What does "forest" mean here? Fruit trees? Please specify.

Authors' response:
The forest in the study area are a mixture of coniferous and deciduous trees. Forest should in the context of the study area not fall under agriculture. Also based on the suggestion by Referee 1, forest and heath are removed from the sentence. The first sentence of the paragraph has been modified to include this information.

From the 2015 land use map from Statistics Netherlands can be deduced that 70.2 % of the land is used for agricultural activities, 13 % is mixed coniferous and deciduous forests, 11.3 % is built-up and the remaining 5.5 % is classified as water, recreational, dry and wet nature.

**R2SC3:**
L116. Probably harvested in September and October.

Authors' response:
Only in the recent dry years farmers started with the harvest maize in September. In the past it was not unusual that the harvest would be postponed to November because the maize cobs needed more time to mature. The text is modified as follows:

Maize is planted in the months April/May and harvested in the period from September to November depending on the vehicle bearing capacity of the land and growing conditions.

**R2SC4:**
L185. What has happened in case of sensor failure? What if sensors had to be replaced? Was the same sensor type used? Was there a check for inhomogeneity in the measurement time series?

Authors' response:
In the case of sensor failure the sensor was replaced by a similar METER group sensor, type EC-TM or 5TM. Prior to installation, every sensor was tested for its functionality using measurements of air and water.

The sensor-to-sensor variability was accounted for by the manufacturer's sensor calibration against known dielectric standards as we wrote on l192-194. The installed sensor type is included in the data quality flags as explained on l333-337. This issue with the internal calibration of a specific batch of the 5TM sensors has been dealt with as reported in l354-359.

The following sentence has been added to the first paragraph of section 3.2.

The functionality of the probes were tested using measurements of water and air prior to deployment and the installed probe types are documented as a quality flag within the datasets, see section 6.

**R2SC5:**
L296-L303. Can you explain why these RMSE´s are so large? What is the RMSE for the average soil moisture content of a complete field or area?

Authors' response:
The RMSE values are not specifically large for field campaigns during which soil moisture is measured in multiple fields and over the full dynamic range. The uncertainty levels are generally lower under controlled laboratory conditions or for a single prepared field because under those conditions soil sampling is more reliable due to the absence of natural variabilities in, for instance, soil clods and plant roots. This explanation is provided in the context of the conducted field campaigns around l300-303.

For comparison, Cosh et al. (2005) report on RMSEs varying from 0.027 up to 0.041 $m^3$ $m^{-3}$ and from 0.040 $m^3$ $m^{-3}$ up to 0.054 $m^3$ $m^{-3}$ for field specific and soil specific calibrations, respectively, see copied table below. We obtain for the 2009 campaign the highest RMSE value of 0.048 $m^3$ $m^{-3}$. This field campaign had the lowest number of sampling days in combination with the largest number fields sampled, which may have led to a larger uncertainty. This explanation is given in l271-274 of the manuscript. We obtain for the 2015 field campaign a measurements uncertainty of 0.041 $m^3$ $m^{-3}$ with the ThetaProbe and for the 2016/17 campaign an overall RMSE of 0.032 $m^3$ $m^{-3}$ is achieved with the HydraProbe. So all-in-all the RMSEs are in line with the reported state-of-art.

Table 2
Summary statistics for the SMEX02 impedance probe calibration methods

| Data set | Generalized calibration | Soil specific calibration | Field specific calibration |
| --- | --- | --- | --- |
| WC region | | | |
| $R^2$ | 0.698 | 0.698 | 0.787 |
| Bias ($m^3/m^3$) | 0.022 | 0.001 | 0.000 |
| rmse ($m^3/m^3$) | 0.061 | 0.049 | 0.041 |
| IA region | | | |
| $R^2$ | 0.744 | 0.742 | 0.803 |
| Bias ($m^3/m^3$) | 0.009 | −0.014 | 0.000 |
| rmse ($m^3/m^3$) | 0.053 | 0.054 | 0.040 |
| Little Washita (LW) | | | |
| $R^2$ | 0.367 | 0.370 | 0.612 |
| Bias ($m^3/m^3$) | −0.010 | −0.006 | 0.001 |
| rmse ($m^3/m^3$) | 0.057 | 0.051 | 0.039 |
| OS region | | | |
| $R^2$ | 0.713 | 0.722 | 0.844 |
| Bias ($m^3/m^3$) | 0.013 | 0.014 | 0.000 |
| rmse ($m^3/m^3$) | 0.039 | 0.040 | 0.027 |
| ON region | | | |
| $R^2$ | 0.571 | 0.571 | 0.760 |
| Bias ($m^3/m^3$) | 0.003 | 0.007 | −0.001 |
| rmse ($m^3/m^3$) | 0.048 | 0.040 | 0.028 |
| Total | | | |
| $R^2$ | 0.716 | 0.716 | 0.821 |
| Bias ($m^3/m^3$) | 0.001 | −0.006 | 0.000 |
| rmse ($m^3/m^3$) | 0.053 | 0.050 | 0.037 |

Table copied from Cosh et al. (2005)

Cosh, M.H., Jackson, T.J., Bindlish, R., Famiglietti, J.S., Ryu, D.: Calibration of an impedance probe for estimation of surface soil water content over large regions, Journal of Hydrology, 311, 49-58, doi:10.1016/j.jhydrol.2005.01.003, 2005.

**R2SC6:**
L335. iv) instead of v)

Authors' response:
done

**R2SC7:**
L343: Change to: "a readme document"

Authors' response:
We have changed this to 'the readme document accompanying the dataset'.

**R2SC8:**
L366. Do you compare here individual measurement points with measurements?

Authors' response:
These error metrics are based on the comparisons of the stations with the field average soil moisture obtained from the campaigns. This issue will be clarified in the new section on spatial representativeness that was suggested by referee 1.

**R2SC9:**
L375. "lower groundwater levels" instead of "low groundwater levels"?

Authors' response:
Done.

**R2SC10:**
L391. How doe you explain this? Could it be related to preferential flow in the unsaturated zone?

Authors' response:
It is possible that preferential flow is an explanation. Another more likely explanation is that the shallow groundwater table causes a naturally fast response of the surrounding hydrology, especially in winters. This leads to the fact that groundwater table fluctuations match relatively well with temporal variations in soil moisture measured at 5 cm and 10 cm measured. The moisture content measured at 40 and 80 cm is under those conditions less responsive to rain events because the surrounding soil is already saturated. The following three sentences are added.

This can likely to be attributed to the shallow groundwater table in the study area that causes a naturally fast hydrological response. The groundwater table fluctuations match especially in winters well with the variations in soil moisture measured at 5 cm and 10 cm. The moisture contents measured at 40 and 80 cm is under those conditions less responsive to rain events because the surrounding soil is already saturated.

**R2SC11:**
L394. Or opposite? In situ groundwater levels (whose availability is more abundant than soil moisture measurements) provide information on soil moisture content.

Authors' response:

Indeed, in the western world in-situ groundwater measurements often (readily) available and soil moisture measurements not. In the majority world, however, in-situ groundwater monitoring networks are seldomly in place, while societies have a great demand for them. The text is modified as follows,

Hence, there lies also an opportunity to further investigate the connection between the water content in the unsaturated zone and the groundwater table. This knowledge may be used to provide soil moisture estimates in regions where groundwater monitoring wells are abundant or groundwater information based on surface soil moisture observed from space in countries where groundwater monitoring networks are absent.

**R2SC12:**
L423. Typo: "third-party".

Authors' response:
done

**R2SC13:**
Figure 8. Please explain the numbers in the figure (why twice "1", "2", "3" etc.)

Authors' response:
These numbers stand for the measurement locations within a unique field. We have decided to remove this figure based on the suggestion by referee 1 and because we find it in retrospect of little added value to the text.

---

## Referee Report (RR1)

**Review on essd-2022-90 by van der Velde an colleagues**

I would like to thank the authors for addressing and responding to the comments.

I still have some comments which should be addressed before publication. Many of these comments apply to the new sections 5.2 and 6 which are interesting, but still a bit unwieldy. I would like to ask the authors to carefully go through these sections once more. I understand that with my comments in the interactive discussion, I motivated the authors to put more emphasis on these parts, and I appreciate the reaction. But as these sections are more oriented towards scientific interpretation (instead of the description of the dataset), we have to be more careful and rigorous with regard to formulating hypotheses. I also found one or two issues that were already present in the original preprint, but which slipped my attention. I apologise, but would still ask to address these issues as well.

ll. 18-20: Without reading section 5.2, these lines cannot be understood: How can the spatial representativeness be measured by the $R^2$ or RMSE? What is meant by network scale? Please find a more concise way to summarise your findings on representativeness in the abstract.

l. 21: VSM - acronym not explained in the abstract.

ll. 18-25: Overall, I find these newly added lines difficult to read. Please try to make this more concise.

ll. 147-148: "[...] while typically less than 50 mm were recorded per day." Given that the most extreme daily rainfall depths were reported as 50, 142 and 106 mm, it is pretty obvious that the other days had less rainfall. Hence, his fragment does not bear any information. Please delete.

l. 156 should read "In the site selection, care was taken to evenly distribute the SENSOR LOCATIONS across [...]"

l. 173: should be "section 7", now, I suppose.

l. 180: Why "soil layer" instead of just "soil"?

l. 183-184: Where can I find the information which locations have a limited coverage of measurement depths? Should this information go into Tab. S2?

ll. 211-213: I suggest using standard terminology to refer to this procedure (leave-one-out cross validation).

l. 227: I suggest using "surface soil moisture" and also label the section "Field campaigns to observe surface soil moisture", so that it becomes clearer to the reader that this is not about SWC profiles.

l. 289-290: "agreement difference" sounds weird. I suggest to replace the entire sentence "Factors that could have contributed to this agreement difference are the deployed instruments [...]" by "This could be explained by the deployed instruments, [....]"

Section 5.2: In the beginning of this section, you should again highlight that any of the following analysis only refers to the agreement at the upper 5 cm. It does not tell us anything about what's happening below (in terms of representativeness).

l. 336: you replaced "representativeness for the field" by "representativeness of the field" which is not correct, in my opinion.

ll. 337-339: the factors you mention here apply to most soil moisture measurements. What is most important, in my view, is that the soil management between the fields is different from in the field, namely that the fields are usually ploughed and harrowed while the stripes inbetween remain undisturbed. This might have fundamental implications for soil hydraulic properties in the upper 30 cm. Please discuss this briefly, if you agree.

l. 347: "which can be attributed to edge effects": this is just a hypothesis, so I suggest not to make the statement that absolute.

ll. 349-350: not only higher interception losses, but also higher transpiration, wouldn't you agree?

l. 351: "majority" - why so unspecific? Couldn't you just state the number of profiles which fall into that range?

l. 361: "this may be argued for" - please rephrase

l. 364: I would not use "performance", but rather "agreement"

l. 365: "antecedent precipitation" - antecedent over which period before the campaigns?

l. 366: But did you systematically sample, within the field, in local depressions? Otherwise, this would not explain the systematic underestimation, right?

l. 377: "inflated" is not an adequate term, here. Use "high" or "large" instead (if you in fact think it is large).

Fig. 8 and section 6.1:

- I appreciate the motivation to combine the campaign measurements with the continuous measurements. Still, I am having some difficulties to understand the figure and its purpose. The red lines represent the "network", so all soil moisture profiles in Twente? But averaged over all depths? Or just at the surface (upper 5 cm)? And the markers represent any profile/field for which a campaign was carried out on a given day? This needs better explanation.

- Apart from comprehensibility, what can we actually learn from contrasting the means of selected subsets of the data with the overall network mean? You state that the figure reflects the network's "overall performance" - what is meant by that?
- On what basis do you state that the campaigns measurements match the station measurements "very well" (l. 398).
- ll. 403 ff.: I am quite hesitant about the presented concept of "temporal representativeness": "[we] found that the least differences between the values measured during the field campaigns and stations' data records do not necessarily occur at the same time of measurement." To be honest, I do not understand what is implied here and which "physical processes" you refer to. I am not doubting the stated fact, but I am wondering about any explanation beyond "random effect". Please elaborate.
- Technical remarks: (i) do not use filled markers, but wider edgelines for the markers instead. (ii) And which precipitation observations are shown on the secondary axis? Or is this an average of all rain gauges? If yes, how is it averaged/weighted? (iii) The first legend should use four columns and one row instead of one column and four rows.

Fig. 9: In order to adequately interpret the figure, precipitation and air temperature need to be shown, too, on secondary axes and/or an additional panel.

Fig. 10:
- To better understand the effect atmospheric drivers, I usually find it helpful to display the cumulative sum of the daily difference between precipitation and reference evapotranspiration. That way, you can typically see a clear relationship between increasing and decreasing parts of that curve and the drying and wetting of the topsoil. This is just a suggestion, since showing daily air temperature and precipitation for such long time series is difficult to interpret.
- How can the volumetric SWC be higher than 0.6 m³/m³, even close to 0.8 m³/m³ on a location with sand / highly loamy sand (ITC_SM14, see Tab. S2, and also ITC_SM17). I find this quite spurious.

ll. 430: Instead of "Specifically in the 80 cm soil moisture content [...]" better "Specifically at a depth of 80 cm, soil moisture content [...]"

l. 434: replace "measurement" by "level" and "increments" by "increases"

ll. 435-438: In my view, care needs to be taken with such correlations. I understand that this is just a data description paper, so that in-depth analyses are unwarranted. Yet, when correlating SWC time series with the groundwater level, the delays between the signals, corresponding to the travel of the water from the soil down to the aquifer, should be taken into account. At least for many soils without predominant bypass flow, there should be such a clear delay. Before correlating the time series in order to identify which SWC signal best explains groundwater level dynamics, the delay should be accounted for by shifting the series in time, finding the shift which yields the maximum correlation. As I guess such an analysis is taking things too far for this paper, the authors might consider removing the table and the corresponding text fragments.

ll. 450-452: It is unclear what the authors mean by "[...] make it possible to address sub-catchment scale applications."

ll. 498: You use the term "network scale" which I find insufficiently defined. Maybe rather "for the entire network" if you refer to specific statistical metrics.

ll. 502: To what does the "hence" refer?

ll. 505: "network scale" - see above.

ll. 505-508: How can your dataset be valuable for upcoming (future) SAR missions if it only spans until 2020??

Please provide all figures (except 1 and 3) in vector format in order to allow for lossless zooming.

Check for consistent use of tense throughout the manuscript, specifically in the newly added parts.

---

## Author Response (AR2)

Dear Editor(s) of the Earth System Science Data (ESSD) journal,

Dear Dr. Sibylle Hassler,

We would like to thank you for considering our manuscript 'Twelve years profile soil moisture and temperature measurements in Twente, The Netherlands' (essd-2022-90) for publication in the well-established ESSD journal and offering for the second time the opportunity to revise and resubmit it. Included in this submission is a clean copy of the revised manuscript, a marked-up version of the manuscript showing the changes made and detailed point-by-point responses to the comments of referee 1.

We have carefully considered the comments, questions, and suggestions for improvement made by referee 1 and applied them where we found appropriate. The major changes can be summarized as follows:

- Textual improvements have been made throughout the manuscript.
- The text with respect to analysis of the spatial representativeness has been updated to emphasize that the representativeness of the top 5 cm soil moisture is discussed.
- The second paragraph of the abstract has been rewritten to make better readable standalone.
- Figures 8, 9 and 10 have been updated according to the referee's suggestions.

The comments of the referee 1 have again helped us to improve the overall quality of the manuscript, for which we are grateful.

Overall we trust that we have addressed all the referee comments, and have further improved the quality of the manuscript to the level expected from publications in the ESSD journal. We would like to thank you for taking the time to handle this contribution and look forward to hearing from you.

Yours truly,

Rogier van der Velde
On behalf of the authors.

**Authors' response to Referee 1**

I would like to thank the authors for addressing and responding to the comments.

I still have some comments which should be addressed before publication. Many of these comments apply to the new sections 5.2 and 6 which are interesting, but still a bit unwieldy. I would like to ask the authors to carefully go through these sections once more. I understand that with my comments in the interactive discussion, I motivated the authors to put more emphasis on these parts, and I appreciate the reaction. But as these sections are more oriented towards scientific interpretation (instead of the description of the dataset), we have to be more careful and rigorous with regard to formulating hypotheses. I also found one or two issues that were already present in the original preprint, but which slipped my attention. I apologise, but would still ask to address these issues as well.

Authors' response:
We thank the referee for the positive and detailed comments. Again the level of detail and provided suggestions are much appreciated. In the text below, we provide our point-by-point response to the comments.

The referee's suggestions have been implemented in the revised manuscript. Our response to the referee's comments is structured as follows:
- The native referee comment is labelled and written in black.
- The authors response is written in blue
- Text from the manuscript is written with the Times New Roman font whereby the native text is in black and the changes are in red.
- The line numbers refer to the clean copy of the revised manuscript.

**Comment R1C1**
ll. 18-20: Without reading section 5.2, these lines cannot be understood: How can the spatial representativeness be measured by the $R^2$ or RMSE? What is meant by network scale? Please find a more concise way to summarise your findings on representativeness in the abstract.

Authors' response:
We removed the reference to specific $R^2$ and RMSE values, avoided the use of the words 'network-scale' and summarise the main findings more precise and in a more concise manner.

**Comment R1C2**
l. 21: VSM - acronym not explained in the abstract.

Authors' response:
We replaced VSM by field averaged soil moisture content

**Comment R1C3**
ll. 18-25: Overall, I find these newly added lines difficult to read. Please try to make this more concise.

Authors' response:
We have rewritten the text as follows,

L18-24:
An indication for the spatial representativeness of the permanent monitoring stations is provided through comparisons of the 5 cm station measurements with the top 5 cm field averaged soil moisture content derived from the field campaign measurements. The results reveal in general reasonable agreements and root mean squared errors that are dominated by underestimations of the field averaged soil moisture content, which is particularly apparent for the grass fields and strong after heavy rain. Further, we discuss the prospects the datasets offer to investigate i) the reliability of soil moisture references that serve the development and validation of soil moisture products, and ii) the water and energy exchanges across the groundwater-vadose zone – atmosphere continuum within a lowland environment in a changing climate.

**Comment R1C4**
ll. 147-148: "[…] while typically less than 50 mm were recorded per day." Given that the most extreme daily rainfall depths were reported as 50, 142 and 106 mm, it is pretty obvious that the other days had less rainfall. Hence, his fragment does not bear any information. Please delete.

Authors' response:
done

**Comment R1C5**
l. 156 should read "In the site selection, care was taken to evenly distribute the SENSOR LOCATIONS across […]"

Authors' response:
We changed this to monitoring locations to be consistent with the used terminology.

**Comment R1C6**
l. 173: should be "section 7", now, I suppose.

Authors' response:
done

**Comment R1C6**
l. 180: Why "soil layer" instead of just "soil"?

Authors' response:
We changed this to soil-water-air mixture

**Comment R1C7**
l. 183-184: Where can I find the information which locations have a limited coverage of measurement depths? Should this information go into Tab. S2?

Authors' response:
We have added this information to a newly created Table S3 and added to the main body of the manuscript the following:

L182:
Table S3 provides for each station the installed sensor types and installation depths.

**Comment R1C8**

ll. 211-213: I suggest using standard terminology to refer to this procedure (leave-one-out cross validation).

Authors' response:
The two sentences have been replaced by

L210-211:
The leave-one-out cross-validation procedure is adopted for calculating the performance metrics because of the limited sample size and to provide an uncertainty estimate for coefficients $a$ and $b$.

**Comment R1C9**
l. 227: I suggest using "surface soil moisture" and also label the section "Field campaigns to observe surface soil moisture", so that it becomes clearer to the reader that this is not about SWC profiles.

Authors' response:
Done. We also mention in the abstract that the top 5 cm soil moisture content is measured during the field campaigns, and swapped and rewrote the second sentence at the start of section 4 with the first sentence of section 4.1 as follows:

L225-227:
The objective of the campaigns was the validation soil moisture retrievals from satellite observations via estimates of the spatially aggregated top 5 cm soil moisture content, hereafter referred to as surface soil moisture.

L234:
Sampling took place at up to five fields near a monitoring station with in total of 28 sampled fields near 12 monitoring stations.

**Comment R1C10**
l. 289-290: "agreement difference" sounds weird. I suggest to replace the entire sentence "Factors that could have contributed to this agreement difference are the deployed instruments […]" by "This could be explained by the deployed instruments, [….]"

Authors' response:
done

**Comment R1C11**
Section 5.2: In the beginning of this section, you should again highlight that any of the following analysis only refers to the agreement at the upper 5 cm. It does not tell us anything about what's happening below (in terms of representativeness).

Authors' response:
We renamed section 5.2:
Spatial representativeness of observed surface soil moisture.

Rewritten the sentence around L331-332 as follows:
Field averages derived from the surface soil moisture measurements collected during the campaigns (see section 4) have been used to assess this issue.

And refer to 'surface soil moisture' in various parts of this section.

**Comment R1C12**
l. 336: you replaced "representativeness for the field" by "representativeness of the field" which is not correct, in my opinion.

Authors' response:
We have changed this to, its representativeness for the field.

**Comment R1C13**
ll. 337-339: the factors you mention here apply to most soil moisture measurements. What is most important, in my view, is that the soil management between the fields is different from in the field, namely that the fields are usually ploughed and harrowed while the stripes in between remain undisturbed. This might have fundamental implications for soil hydraulic properties in the upper 30 cm. Please discuss this briefly, if you agree.

Authors' response:
To address this issue, we have modified the sentence as follows,

L328-332:
Large differences in the meteorological inputs, e.g. precipitation and incoming solar radiation, are not expected, but small-scale topography, spatially variable soil texture, different land covers and degrees of soil compaction as a result of agricultural management practices, and field-specific drainage infrastructure may cause discrepancies between the VSM at the border and inside of the field.

**Comment R1C14**
l. 347: "which can be attributed to edge effects": this is just a hypothesis, so I suggest not to make the statement that absolute.

Authors' response:
We have changed the word 'can' to 'may '.

**Comment R1C15**
ll. 349-350: not only higher interception losses, but also higher transpiration, wouldn't you agree?

Authors' response:
We agree and changed this to 'transpiration and interception of precipitation'

**Comment R1C16**
l. 351: "majority" - why so unspecific? Couldn't you just state the number of profiles which fall into that range?

Authors' response:
We have modified the sentence as follows,

L341-342:
The $R^2$ values range for six out of the nine stations from 0.516 to 0.793, while $R^2$ values of 0.36 and 0.38 suggest that stations ITCSM_05 and ITCSM_18 are less representative of the fields.

**Comment R1C17**
l. 361: "this may be argued for" - please rephrase

Authors' response:

We have replaced this may be argued for by this may be explained by

**Comment R1C18**
l. 364: I would not use "performance", but rather "agreement"

Authors' response:
done

**Comment R1C19**
l. 365: "antecedent precipitation" - antecedent over which period before the campaigns?

Authors' response:
We have modified the sentence as follows,

L352-354:
Further analysis shows that a large part of this weaker agreement stems from two days (19 October 2016 and 28 June 2017) with exceptionally large mismatches, which have in common that on average more than 27 mm of precipitation was recorded in total on the day itself and the day before.

As the text states on average more the 27 mm of precipitation was recorded on average by summing the volume of the day itself and the before. The 27 mm is derived as the average of those sums recorded by the three automated weather stations. The temporal precipitation distribution are a little bit different for the two days. Prior to 18/19 October 2016 it was relatively dry and the majority of the rain fell on October 19. We took the field measurements during/after a down poor. Prior to 27/28 June 2017 it had rained already on five consecutive days and on 27/28 extra rain fell.

For the sake of brevity, we did not include this detailed explanation because we find that this information does not add to the narrative.

**Comment R1C20**
l. 366: But did you systematically sample, within the field, in local depressions? Otherwise, this would not explain the systematic underestimation, right?

Authors' response:

It is true that the Netherlands is flat but it is not as flat as a pancake as people may think. We sampled during those day and I (the first author) remember very well that parts of the field were completely submerged whereas other parts remained clear of standing water.

To clarify this, we have modified the sentence as follows,

L356-358:
The large rain volumes on those days most likely led to overland flow that accumulates in local depressions and led on those days to the partial flooding of fields as a result of small scale topography, whereas the instrumentation at ITCSM_10 was installed in a slightly higher and, therefore, drier part of the field.

**Comment R1C21**
l. 377: "inflated" is not an adequate term, here. Use "high" or "large" instead (if you in fact think it is large).

Authors' response:
We replaced 'inflated' by 'large'.

**Comment R1C22**
Fig. 8 and section 6.1:
- I appreciate the motivation to combine the campaign measurements with the continuous measurements. Still, I am having some difficulties to understand the figure and its purpose. The red lines represent the "network", so all soil moisture profiles in Twente? But averaged over all depths? Or just at the surface (upper 5 cm)? And the markers represent any profile/field for which a campaign was carried out on a given day? This needs better explanation.
- Apart from comprehensibility, what can we actually learn from contrasting the means of selected subsets of the data with the overall network mean? You state that the figure reflects the network's "overall performance" - what is meant by that?

Authors' response:
The purpose of the figure is to evaluate how the biases for individual fields propagate when aggregated over a number of fields. This provides a best estimate for the bias of the entire network since we have not been in the position to sample near all our monitoring stations on any day in a systematic manner. The mean and standard deviation of the 5 cm VSM of the entire network are shown in figure 8 to illustrate how representative the sampled fields are for the mean 5 cm VSM measured by the entire network. The text has been rewritten as follows,

L360-364:
The metrics labelled 'sampling day' are based on matchups between the mean values of all field-averaged surface soil moisture and corresponding 5 cm station VSM measurements collected on a specific day. They show how the biases found for individual fields propagate when aggregated over a number of fields and provide an indication for the bias of the entire network. In support, Fig. 8 shows the mean of the field-averaged surface soil moisture and the matching mean of the 5 cm station VSM for the years 2015, 2016 and 2017 along with the mean 5 cm VSM of the entire network plus and minus the standard deviation.

- On what basis do you state that the campaigns measurements match the station measurements "very well" (l. 398).

Authors' response:
We make this statement based on the $R^2$ value of 0.770 reported in section 5.2. In analogy with the terminology used in that section, we scale 'very well' down to 'fairly well' and clarified the context by rewriting the text as follows,

L390-394:
Figure 8 shows that the mean values of all field averaged surface soil moisture and corresponding 5 cm station VSM measurements collected on a specific campaign day match fairly well with each other given a $R^2$ of 0.770 as well as the mean of 5 cm station VSM of the entire network (solid lines). These results provide an indication for the bias of the entire network, but the results presented in section 5.2 also demonstrate that further investigations should address the effect of spatial heterogeneity at field-scale.

- ll. 403 ff.: I am quite hesitant about the presented concept of "temporal representativeness": "[we] found that the least differences between the values measured during the field campaigns and stations' data records do not necessarily occur at the same time of measurement." To be honest, I do not understand what is implied here and which "physical processes" you refer to. I am not doubting the stated fact, but I am wondering about any explanation beyond "random effect". Please elaborate.

Authors' response:
We agree that the term 'temporal representativeness' is not appropriate. We refer here to a mismatch between the station VSM measured with probes installed at a depth of 5 cm and the top 5 cm soil moisture (surface soil moisture) measured during the field campaigns through soil sampling and with impedance probes.

We have rewritten the text as follows,

L396-401:
In addition, the presented data enables research into the representativeness of station VSM measured with probes installed at a depth of 5 cm for the top 5 cm soil moisture measured during campaigns that are typically considered as reference in validation studies. We carried out a preliminary analysis and found that the best match between the surface soil moisture measured during the field campaigns is found with the 5 cm station VSM that is recorded several hours up to two days later. The presented datasets provide an opportunity to investigate this and the physical processes that affect the near surface soil moisture profile, in particular infiltration and evaporation.

- Technical remarks: (i) do not use filled markers, but wider edgelines for the markers instead. (ii) And which precipitation observations are shown on the secondary axis? Or is this an average of all rain gauges? If yes, how is it averaged/weighted? (iii) The first legend should use four columns and one row instead of one column and four rows.

Authors' response:

The figure has been modified according to the reviewer's suggestions and the caption is updated to explain the precipitation data source:

[Figure]

**Figure 8: Mean values of field-averaged surface soil moisture measured during the 2015, 2016, and 2017 field campaigns (marker: circle) and of 5 cm VSM measured at the matching monitoring stations (marker: square). The solid and dotted lines represent the mean 5 cm VSM of the entire network +/- the standard deviation. The precipitation shown on the secondary axis is derived as the arithmetic mean from the data collected by the three KNMI AWSs.**

**Comment R1C23**

Fig. 9: In order to adequately interpret the figure, precipitation and air temperature need to be shown, too, on secondary axes and/or an additional panel.

Authors' response:

We update Figure 9 by adding a new panel depicting hourly values of air temperature and precipitation measured at the KNMI AWS Twenthe airport

This figure displays the nexus between precipitation-soil moisture and temperature. For example, the rainfall events spread over 11 and 13 July had a peak intensity of 3 mm h$^{-1}$ at an average air temperature of 16.5 °C and showed an effect on the 5, 10 and 20 cm soil moisture content. The following days a smaller amplitude in the diurnal air and soil temperature cycle can be noted and the soil dried out gradually. The rainfall event of 20 July had a lower peak intensity of 2 mm h$^{-1}$ and fell when the air temperature was higher, on average 19.9 °C, and did not led to a change in the soil moisture contents

[Figure]

**Figure 9: Soil moisture and temperature depth profiles measured at ITC_SM07 from 7 till 31 July covering a 2019 heatwave in Northwestern Europe. The upper panel shows the hourly precipitation and air temperature measured the KNMI AWS Twenthe airport about 12 km southwest of ITC_SM07.**

**Comment R1C24**

Fig. 10:

- To better understand the effect atmospheric drivers, I usually find it helpful to display the cumulative sum of the daily difference between precipitation and reference evapotranspiration. That way, you can typically see a clear relationship between increasing and decreasing parts of that curve and the drying and wetting of the topsoil. This is just a suggestion, since showing daily air temperature and precipitation for such long time series is difficult to interpret.
- How can the volumetric SWC be higher than 0.6 m³/m³, even close to 0.8 m³/m³ on a location with sand / highly loamy sand (ITC_SM14, see Tab. S2, and also ITC_SM17). I find this quite spurious.

Authors' response:

For the revised manuscript we decided to plot the daily precipitation minus the reference evapotranspiration instead of the accumulations as the referee suggests. We find that the daily precipitation surplus provides a better indication for period when we may expect a rise in soil moisture content in comparison to the accumulations.

With respect to the second point the referee made, we found that it was based on an earlier version of the dataset that was based on old calibration. For the figure in the revised manuscript, we have used the published calibrated datasets as is, which include significantly lower the soil moisture content.

Still, high soil moisture values are found, but those are from the Winter/Spring periods of the top 5 cm and 10 cm. Under those conditions, the soil will be completely saturated possibly even with standing water. Moreover, the fields where these high soil moisture values are measured in grasslands that have a high root density and organic matter content near the soil surface. This is not considered in the classification map from which texture class reported in Table S2 has been derived. The following sentence has been added to section 3.1,

L154-155:
It should be noted that near the surface the organic matter content is higher than one would expect based on the texture class and that grasslands have a dense rooting system.

The modified figure is shown on the following page.

[Figure]

**Comment R1C25**

ll. 430: Instead of "Specifically in the 80 cm soil moisture content [...]" better "Specifically at a depth of 80 cm, soil moisture content [...]"

Authors' response:
We have changed the text as follows,

Specifically at a depth of 80 cm, the effects of 2018, 2019 and 2020 droughts are visible, while …

**Comment R1C26**
l. 434: replace "measurement" by "level" and "increments" by "increases"

Authors' response:
done

**Comment R1C27**
ll. 435-438: In my view, care needs to be taken with such correlations. I understand that this is just a data description paper, so that in-depth analyses are unwarranted. Yet, when correlating SWC time series with the groundwater level, the delays between the signals, corresponding to the travel of the water from the soil down to the aquifer, should be taken into account. At least for many soils without predominant bypass flow, there should be such a clear delay. Before correlating the time series in order to identify which SWC signal best explains groundwater level dynamics, the delay should be accounted for by shifting the series in time, finding the shift which yields the maximum correlation. As I guess such an analysis is taking things too far for this paper, the authors might consider removing the table and the corresponding text fragments.

Authors' response:
We agree that assuming a linear relationship between soil moisture and groundwater is an oversimplification and not suitable to draw conclusions. However, a more detailed analysis, as the referee suggests, goes indeed beyond the scope of this article. We have, therefore, remove the table and corresponding text.

**Comment R1C28**
ll. 450-452: It is unclear what the authors mean by "[…] make it possible to address sub-catchment scale applications."

Authors' response:
We have modified the text as follows:

L443:
make it possible to study smaller scale applications than were addressed before.

**Comment R1C29**
ll. 498: You use the term "network scale" which I find insufficiently defined. Maybe rather "for the entire network" if you refer to specific statistical metrics.

Authors' response:
As suggested we have replaced 'network scale' by 'for the entire network'

**Comment R1C30**
ll. 502: To what does the "hence" refer?

Authors' response:
We have removed the word 'hence' and let this sentence be the start of a new paragraph.

**Comment R1C31**

ll. 505: "network scale" - see above.

Authors' response:

We have replaced 'field and network scale' with 'footprints and grid cells' to connect with the satellite and model-based products referred to earlier in the sentence. The sentence is modified to read,

L494-495:

… to develop and validate satellite and model-based soil moisture products at the scale of footprints and grid cells.

**Comment R1C32**

ll. 505-508: How can your dataset be valuable for upcoming (future) SAR missions if it only spans until 2020??

Authors' response:

We intended to convey that new insights into the uncertainties of soil moisture reference may be helpful for the development of calibration/validation plans of upcoming satellite missions, such as the upcoming NISAR and ROSE-L mission. We have modified the text as follows,

L495-498:

This may, for instance, be relevant for the development of calibration/validation plans for upcoming Synthetic Aperture Radar (SAR) missions such as the NASA-ISRO SAR mission (NISAR; Kellogg et al., 2020) and the Radar Observing System of Europe L-band (ROSE-L; Davidson & Furnell, 2021), which have both soil moisture included as part of their mission objectives.

**Comment R1C33**

Please provide all figures (except 1 and 3) in vector format in order to allow for lossless zooming.

Authors' response:

We have all figures (except 1 and 3) in .pdf format that will be included in the publication package.

**Comment R1C34**

Check for consistent use of tense throughout the manuscript, specifically in the newly added parts.

Authors' response:

We have checked the entire manuscript and made modifications throughout.

---

## Author Response (AR3)

Dear Editor(s) of the Earth System Science Data (ESSD) journal,

Dear Dr. Sibylle Hassler,

We would like to thank you for considering our manuscript 'Twelve years profile soil moisture and temperature measurements in Twente, The Netherlands' (essd-2022-90) for publication in the ESSD journal and constructive review of the submitted manuscript and dataset. We provide our detailed response to your suggestion on the readme file accompanying the dataset and the organization of the field campaign dataset in the text below.

In response to your suggestion on the 'eye-friendliness' of the readme document we have formatted the text document and transformed it into an OpenOffice document. The revised readme document is included in this submission as a supplement and will be submitted to DANS-easy. For this submission, no changes have been made to the earlier submitted supplement.

With respect to the link between location of the measurements points in Fig. 5 and the measurements in column 1-9, we have modified to following sentences in the readme document with in red the modified text:

> *"Columns 9-17*: Soil moisture probe readings, up to nine, collected at a measurement location. Information on the sampling strategy can be found in section 4.1 of Van der Velde et al. (in review). There is no systematic relationship between the column name and the sampling points indicated in Fig. 5 of the manuscript. Column 29 provides the location of the soil sampling location used for gravimetric determination of the volumetric soil moisture content. Missing data is indicated by -99.999."

Further, the following sentence has been added to the manuscript at lines 246-247:

"Locations of other sampling points are not specified because they were not consistently documented in a consistent manner during the field campaigns."

In brief, the sampling design differed during the field campaigns and were caried out by different sampling teams. The sampling teams had instructions to document the probe reading that matches the soil sample take for gravimetric determination of the volumetric soil moisture content, but did not record each sampling location in a consistent manner during the field campaign. This information is, therefore, not included in the dataset. However, in column 29 of the field campaign dataset we do provide the soil sampling location used for gravimetric determination of the volumetric soil moisture content.

We trust that with the above described modifications to the manuscript and readme document we have clarified your question related to the organization of the field campaign data. We would like to thank you once again for taking the time to handle this contribution and look forward to hearing from you.

Yours truly,

Rogier van der Velde

On behalf of the authors.